# Shu complex SWS1-SWSAP1 promotes early steps in mouse meiotic recombination

Carla M. Abreu [ID] [1], Rohit Prakash [ID] [1], Peter J. Romanienko[2], Ignasi Roig [ID] [3], Scott Keeney [ID] [4] & Maria Jasin [ID] [1]

The DNA-damage repair pathway homologous recombination (HR) requires factors that promote the activity of strand-exchange protein RAD51 and its meiosis-specific homolog DMC1. Here we show that the Shu complex SWS1-SWSAP1, a candidate for one such HR regulator, is dispensable for mouse viability but essential for male and female fertility, promoting the assembly of RAD51 and DMC1 on early meiotic HR intermediates. Only a fraction of mutant meiocytes progress to form crossovers, which are crucial for chromosome segregation, demonstrating crossover homeostasis. Remarkably, loss of the DNA damage checkpoint kinase CHK2 rescues fertility in females without rescuing crossover numbers. Concomitant loss of the BRCA2 C terminus aggravates the meiotic defects in *Swsap1* mutant spermatocytes, suggesting an overlapping role with the Shu complex during meiotic HR. These results demonstrate an essential role for SWS1-SWSAP1 in meiotic progression and emphasize the complex interplay of factors that ensure recombinase function.

[1] Developmental Biology Program, Memorial Sloan Kettering Cancer Center, New York, NY 10065, USA. [2] Genome Editing Core Facility, Rutgers-Cancer Institute of New Jersey, New Brunswick, NJ 08901, USA. [3] Genome Integrity and Instability Group, Institut de Biotecnologia i Biomedicina; Department of Cell Biology, Physiology and Immunology, Cytology and Histology Unit, Universitat Autònoma de Barcelona, Cerdanyola del Vallès, Barcelona 08193, Spain. [4] Molecular Biology Program, Howard Hughes Medical Institute, Memorial Sloan Kettering Cancer Center, New York, NY 10065, USA. These authors contributed equally: Carla M. Abreu, Rohit Prakash Correspondence and requests for materials should be addressed to M.J. (email: m-jasin@ski.mskcc.org)

Maintenance of genomic integrity in mammalian cells requires the regulated assembly, stabilization, and disassembly of RAD51 nucleoprotein filaments for the repair of DNA damage by homologous recombination (HR)[1,2]. Moreover, homology recognition and HR promoted by RAD51 and its meiosis-specific paralog DMC1 ensure faithful homolog pairing and segregation during the first meiotic division for the production of haploid gametes[3,4]. BRCA2 is a multi-domain protein with several interaction sites for both recombinases and promotes and/or stabilizes nucleoprotein filament formation during HR[2,5–8]. Knockout mouse studies have shown that BRCA2 is essential for embryonic development[2], while transgenic mouse studies have demonstrated that BRCA2 is critical for meiotic progression, such that BRCA2 loss in testis causes meiotic arrest and sterility[9]. Like BRCA2, the canonical RAD51 paralogs RAD51B, RAD51C, RAD51D, and XRCC2, which also promote RAD51 activity, are necessary for embryonic development[2], but their role during meiosis is not well understood.

More recently, a new complex, composed of SWS1 and its interacting protein SWSAP1, has also been shown to regulate RAD51 focus formation upon DNA damage, although the reduction in HR with loss of these proteins is not as substantial as that observed with BRCA2 loss[10–12]. Human SWS1 contains a Zn-finger like domain found in the fission yeast Sws1 and the budding yeast Shu2 proteins, although the overall sequence identity is poor ($\leq$20%)[10,13]. SWSAP1 was identified in human cells through its association with SWS1; it is considered to be a novel RAD51 paralog, since like RAD51 and the canonical RAD51 paralogs, it contains Walker A and B motifs which are involved in nucleotide binding/hydrolysis[11]. Unlike SWS1, SWS1-associated proteins are highly diverse in their structure and number in different organisms (i.e., budding yeast, Shu1, Psy3, Csm2; fission yeast Rdl1, Rlp1; worm, RIP-1, and RFS-1)[10,11,14–19]. These proteins are also considered to be RAD51 paralogs, but only some appear to be capable of nucleotide binding/hydrolysis (i.e., human SWSAP1 and worm RFS-1)[11,19], while others are truncated and/or lack the critical Walker motifs[20]. While SWS1 is a defining member of these complexes[10], the variability of the RAD51-like components has led to these complexes being identified as Shu complexes for simplicity, after the budding yeast proteins.

Loss of the Shu complex in diverse organisms reduces HR, but different functional consequences have been observed during meiosis. Fission yeast Shu mutants have normal spore viability[21], but both budding yeast and worm mutants show fertility defects – budding yeast Shu mutants display a 25–50% reduction in spore viability[15,22] and worm Shu mutants show a weak chromosome missegregation phenotype[17–19,23]. In addition, while budding yeast Shu mutants have reduced Rad51 (but not Dmc1) focus formation during meiosis[15], worm mutants have higher numbers of RAD-51 foci, implicating the worm Shu complex in RAD-51 filament disassembly[17,24]. However, in these cases, other markers of meiotic progression have not been examined.

Given the importance of HR in mammalian cells, we knocked out Sws1 and Swsap1 in the mouse. Surprisingly, both mutants are viable. However, they are infertile and show a severe block in meiotic progression associated with reduced RAD51 and DMC1 focus formation. In females, loss of CHK2 partially restores fertility, without restoring normal crossover numbers. In males, concomitant loss of a BRCA2 domain that stabilizes RAD51 filaments leads to an even greater severity in meiotic defects than seen in Shu mutants. These results demonstrate the critical role the Shu complex plays in mammalian meiosis, which can be ameliorated by loss of a DNA damage signaling protein (CHK2) or worsened by loss of a BRCA2 domain (the C terminus).

## Results

**Sws1 and Swsap1 mutant mice are viable but infertile.** To investigate the role of the mouse Shu complex, SWS1-SWSAP1, we disrupted Sws1 (formally Zswim7) or Swsap1 in fertilized eggs using TALE nuclease pairs directed to exon 1 downstream of the translation start site (Fig. 1a, b). From the several mutations obtained (Supplementary Table 1a, b), three frame-shift alleles for Sws1 and two for Swsap1 were selected for further analysis (Fig. 1a, b). Because results are similar for all alleles, most experiments in the main text focus on one mutant for each gene, $Sws1^{\Delta1(A)/\Delta1(A)}$ and $Swsap1^{\Delta131/\Delta131}$, hereafter $Sws1^{-/-}$ and $Swsap1^{-/-}$, unless indicated otherwise in the figure legends. Surprisingly, unlike other RAD51 paralog knockout mice[2], $Sws1^{-/-}$ and $Swsap1^{-/-}$ homozygous animals are viable, as are $Sws1^{-/-}$ $Swsap1^{-/-}$ double mutants (Supplementary Table 2a, b). RT-PCR analysis using testis cDNA derived from $Sws1^{-/-}$ and $Swsap1^{-/-}$ mice confirmed expression of the respective frame-shift alleles (Supplementary Fig. 1a). Mutant mice show no obvious gross morphological defects and have normal body weights (Supplementary Figs. 1b, 2a). However, neither male nor female $Sws1^{-/-}$ and $Swsap1^{-/-}$ mutants are fertile (Supplementary Table 3a). Testis weights from adult single and double mutants are 3- to 4-fold smaller than in control animals, and ovary weights are reduced 3- to 8-fold (Fig. 1c, d; Supplementary Figs. 1b, 2a). Notably, testis weights from mutant juveniles obtained before meiotic arrest has occurred (7.5 days postpartum, dpp) are similar to controls (Fig. 1c).

In testis sections from adult Shu-mutant mice, seminiferous tubules have substantially reduced cellularity and are devoid of post-meiotic germ cells (Fig. 1e and Supplementary Fig. 1c). Spermatocytes appear to arrest during mid-pachynema, possibly at stage IV of the seminiferous epithelial cycle[25]. TdT-mediated dUTP nick end-labeling (TUNEL) demonstrates widespread apoptosis (Supplementary Fig. 2b). Apoptosis in mutant juvenile testes at 7.5 dpp is rarely observed, as in controls, suggesting that pre-meiotic cells are not affected (Supplementary Fig. 2c). Adult ovary sections from mutants lack follicles at any developmental stage (Fig. 1f and Supplementary Fig. 2d). At 3 dpp, ovaries stained for the oocyte marker c-Kit have significantly reduced oocyte numbers in $Sws1^{-/-}$ and $Swsap1^{-/-}$ mice, with some oocytes appearing to be apoptotic (Supplementary Fig. 2e). Together, our data suggest that SWS1 and SWSAP1 are essential for meiotic progression in both male and female mice.

**Meiotic RAD51 and DMC1 focus assembly requires SWS1-SWSAP1.** Testes and ovaries from Shu-mutant mice resemble those of HR- and synapsis-defective mutants, such as $Dmc1^{-/-}$ and $Sycp1^{-/-}$ (ref. [26–29]). To test if the Shu complex is required for HR and/or synapsis, we analyzed the synaptonemal complex (SC), a tripartite proteinaceous structure that forms between the homolog axes as they pair, by immunostaining surface-spread spermatocytes for the SC central region (SYCP1) and axial/lateral elements (SYCP3)[3]. Spermatocytes were also stained for the testis-specific histone H1 variant (H1t), which specifically labels cells at mid-pachynema and beyond[30]. H1t-positive spermatocytes are significantly reduced in mutant testes, indicating an early-pachytene arrest that is bypassed in only a fraction of cells (Fig. 2a, b and Supplementary Fig. 3a, b). The ability of some cells to progress contrasts with $Dmc1^{-/-}$, in which H1t-positive cells are absent[31]. Unlike later stages, the relative proportion of early meiotic prophase cells at leptonema and zygonema are increased in Shu single- and double-mutant mice.

Synaptic abnormalities in the Shu mutants begin to be observed at early zygonema, such that chromosomes are seen with long axes but no synapsis, which become more pronounced

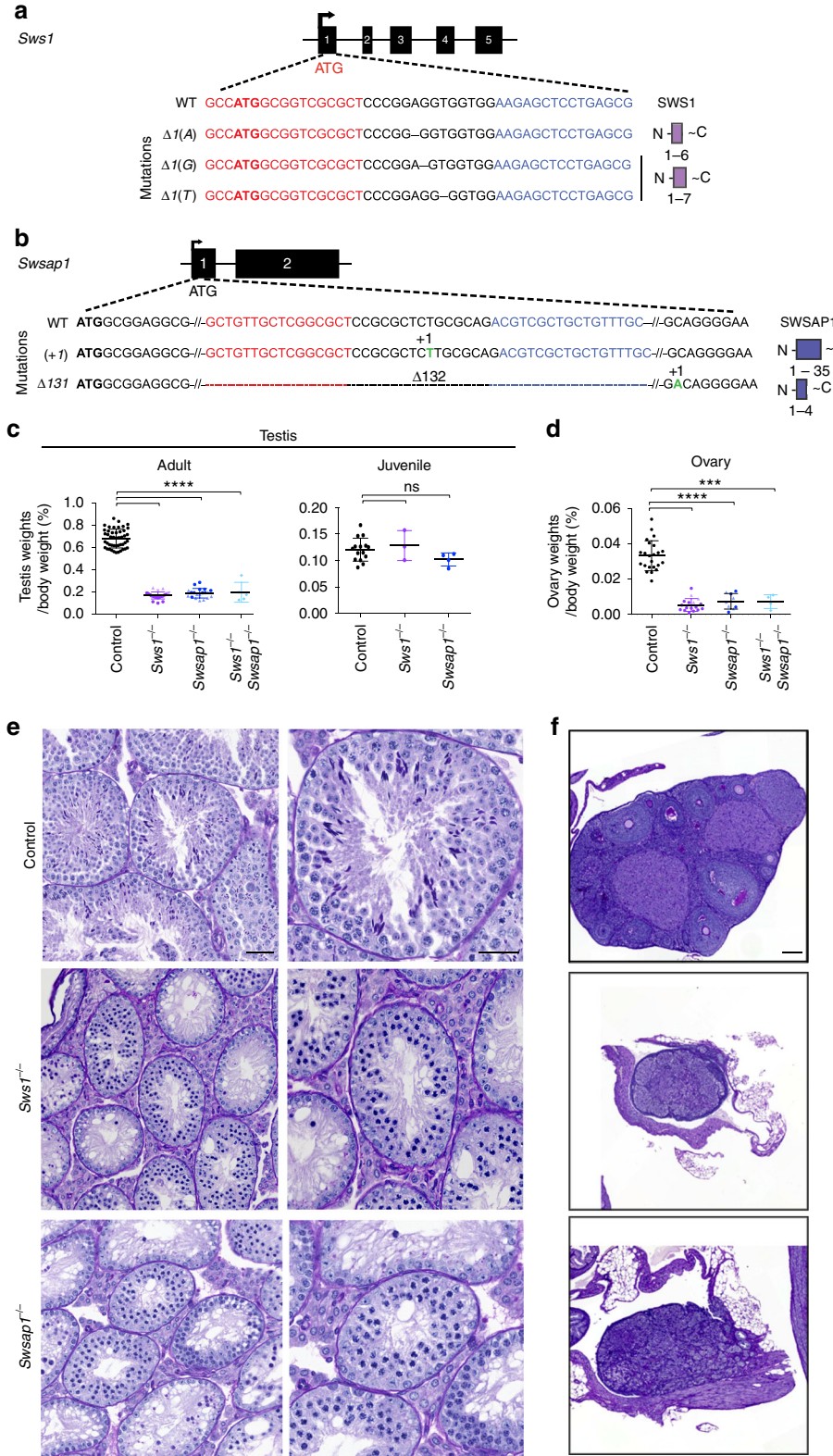

by late zygonema, where chromosomes with fully formed axes are found with little or no synapsis ("early- and late zygonema-like", respectively; Fig. 2b, c and Supplementary Fig. 3b, d). In contrast to wild-type cells at pachynema, in which all of the homologs are typically synapsed, the majority of mutant cells are abnormal at this stage, displaying unsynapsed or partially synapsed chromosomes and frequent synapsis between non-homologous

chromosomes ("pachynema-like"; Fig. 2b, c and Supplementary Fig. 3b–d, e). While severe, the synaptic defects are not as profound as reported for $Dmc1^{-/-}$ mutants[26,27]. Synaptic defects are also observed in the sex chromosomes, with less than half of mutant spermatocytes at early pachynema having synapsed XY pairs (Supplementary Fig. 3c). Mutant cells with autosomal synapsis defects are more likely to also have unsynapsed XY pairs,

**Fig. 1** Defective meiotic progression in *Sws1* and *Swsap1* mutant males and females. **a**, **b** *Sws1* (**a**) and *Swsap1* (**b**) genomic structure, TALEN target sequences (red and blue), mutations analyzed, and the predicted truncated proteins, if translated (shown on the right). **c** Testis to body weight ratios are reduced in adult single and double mutants, but not in juveniles. Error bars, mean ± s.d. Adult mice: control, $n = 45$ (black circles); *Sws1$^{-/-}$* alleles combined [$\Delta 1(A)$, $n = 20$ (purple circles); $\Delta 1(G)$, $n = 4$ (purple triangles); $\Delta 1(T)$, $n = 3$ (purple inverted triangles)], *Swsap1$^{-/-}$* alleles combined [$\Delta 131$, $n = 10$ (dark blue circles); $(+1)$, $n = 8$ (dark blue triangles)], *Sws1$^{-/-}$ Swsap1$^{-/-}$* ($\Delta 1(A) / (+1)$), $n = 5$ (light blue diamonds). Juvenile mice at 7.5 dpp: control, $n = 15$; *Sws1$^{-/-}$*$\Delta 1(A)$, $n = 3$; *Swsap1$^{-/-}$*$\Delta 131$, $n = 4$. ns not significant; ****$P \leq 0.0001$; Student's *t*-test, two-tailed. **d** Ovary to body weight ratios are reduced in adult single and double mutants. Error bars, mean ± s.d. Adult mice: control, $n = 23$; *Sws1$^{-/-}$* [$\Delta 1(A)$, $n = 11$; $\Delta 1(G)$, $n = 3$; $\Delta 1(T)$, $n = 3$], *Swsap1$^{-/-}$* [$\Delta 131$, $n = 4$; $(+1)$, $n = 5$], *Sws1$^{-/-}$ Swsap1$^{-/-}$* ($\Delta 1(A) / (+1)$), $n = 4$. ***$P \leq 0.001$; ****$P \leq 0.0001$; Student's *t*-test, two-tailed. **e** Shu mutant spermatocytes arrest at pachynema. Sections were stained with periodic acid-Schiff (PAS) and counterstained with hematoxylin. Scale bars, 100 μm and 50 μm for left and right columns, respectively. Control, *Sws1$^{-/-}$*, *Swsap1$^{-/-}$*, $n \geq 3$. **f** Ovaries from Shu mutant adults lack follicles. Sections were stained with PAS. Scale bar, 500 μm. Control, *Sws1$^{-/-}$*, *Swsap1$^{-/-}$*, $n \geq 3$

whereas those with full autosomal synapsis typically have synapsed XY pairs. The few cells that reach mid-pachynema tend to have fewer synaptic abnormalities (Fig. 2b and Supplementary Fig. 3b, d). An increase in chromosomal end-to-end fusions in the Shu-mutant spermatocytes relative to controls is also observed (Supplementary Fig. 3f).

Synapsis defects in Shu-mutant spermatocytes could reflect meiotic HR defects, as the human Shu complex promotes HR in cultured cells[10,11]. Indeed, *Sws1$^{-/-}$* and *Swsap1$^{-/-}$* spermatocytes display an ~2-fold reduction in RAD51 and DMC1 focus numbers at leptonema (Fig. 2d–g and Supplementary Fig. 4a, b). RAD51 and DMC1 focus numbers increase substantially by early zygonema in control cells, but remain low in mutant cells (~3-fold lower). At later stages, focus numbers progressively decrease in all genotypes, including in the mutants. However, RAD51 and DMC1 protein levels are similar in controls and mutants (Supplementary Fig. 4c, d). Shu double-mutant spermatocytes have similarly reduced RAD51 and DMC1 focus numbers at all stages (Supplementary Fig. 4a, b). Thus, our data indicate that SWS1 and SWSAP1 are required for normal homolog synapsis and recombinase focus formation during meiosis.

**Resected DNA intermediates increase with SWS1-SWSAP1 loss.** Sterility can occur in mutants where a similar reduction in RAD51 and DMC1 foci is attributable to fewer DSBs[32]. To rule out effects of Shu complex loss on DSB formation and/or their resection, we quantified the signal intensity of chromatin-bound γH2AX[33], a marker for DSBs, and foci of MEIOB, a meiosis-specific, single-stranded DNA (ssDNA)-binding protein[34,35]. At leptonema and early zygonema, γH2AX levels are indistinguishable from controls (Supplementary Fig. 5a, b), suggesting that DSB formation is unaffected. Further, there are more MEIOB foci at these stages in Shu mutant spermatocytes (1.7-fold at leptonema and 1.3-fold at early zygonema; Fig. 3a, b and Supplementary Fig. 5c), indicating an increase in the number of end-resected intermediates. Notably, the increase in MEIOB foci is not as great as in *Dmc1$^{-/-}$* spermatocytes (2.5- and 1.9-fold, respectively). We interpret these findings to indicate that DSBs are formed in normal numbers and are resected, to be initially bound by MEIOB, in *Sws1$^{-/-}$* and *Swsap1$^{-/-}$* mutants, but that the mouse Shu complex fosters the stable assembly of RAD51 and DMC1 nucleoprotein filaments during meiotic HR, which in turn promotes homolog synapsis.

**Crossovers in *Sws1* and *Swsap1* meiocytes that progress.** Given the early meiotic prophase I defects in Shu-mutant spermatocytes, we expected that HR would be impaired later as well. Consistent with defects in DSB repair, mutant spermatocytes display γH2AX on synapsed autosomes, both in early pachytene cells and in early pachytene-like cells with synapsis defects (Supplementary Fig. 5d). γH2AX mostly disappears from autosomes in control cells and remains concentrated in the unsynapsed XY chromatin

forming the sex body[33]. In contrast, autosomal γH2AX in Shu-mutant cells is accompanied by defects in sex body formation/maturation, especially in those cells with a high degree of autosome asynapsis (Supplementary Fig. 5d, e). Interestingly, the rare mid-pachytene Shu-mutant spermatocytes display less autosomal γH2AX than those at early pachynema and occasionally mature sex bodies (Supplementary Fig. 5f), suggesting that some mutant cells do not trigger the pachytene checkpoint due to greater proficiency in DSB repair. However, autosomal γH2AX remnants are still observed in these *Sws1$^{-/-}$* and *Swsap1$^{-/-}$* "escapers", in agreement with evidence suggesting that the pachytene checkpoint tolerates some unrepaired DSBs[36].

Because a small fraction of Shu-mutant spermatocytes are apparently repair-proficient and progress to mid-pachynema, we asked whether mutant cells could form later HR intermediates. MSH4 stabilizes DNA-strand exchange intermediates, some of which will become crossovers, whereas MLH1 foci mark most crossovers[3,37,38]. Mutant spermatocytes have 2- to 3-fold fewer MSH4 foci from early zygonema to early pachynema, proportional to the earlier reduction in the RAD51 and DMC1 foci (Fig. 3c, d). Remarkably, however, MLH1 foci are reduced on average only ~20% in mid-pachytene cells, and the majority of bivalents have at least one MLH1 focus even though most cells have one or more chromosome pairs lacking a focus (Fig. 3e–g). Thus, spermatocytes that progress have a somewhat reduced but still significant number of crossovers, despite the substantially reduced foci of markers of recombination intermediates (RAD51/DMC1 and MSH4).

As Shu-mutant females are sterile, we also tested for evidence of HR defects in oocytes from embryonic day 18.5, when most have entered pachynema (Supplementary Fig. 5g). MLH1 foci are present in *Swsap1$^{-/-}$* oocytes at mid-pachytene, but fewer as in spermatocytes (Fig. 3h, i). Furthermore, most mutant oocytes have ≥1 chromosome pair that is not synapsed and/or lacks an MLH1 focus (Supplementary Fig. 5h). Therefore, crossover defects in females, appear similar to those in males.

**Sws1 Chk2 and Swsap1 Chk2 double mutant females are fertile.** Most Shu mutant oocytes are eliminated within a few days after birth (Supplementary Fig. 2e), resembling other DNA repair-defective mutants like *Dmc1$^{-/-}$* (ref. [29]). Oocyte loss in *Dmc1$^{-/-}$* mice at 3 weeks can be partially rescued by eliminating the DNA damage checkpoint kinase CHK2, although *Dmc1$^{-/-}$ Chk2$^{-/-}$* mice still lack primordial follicles and most oocytes are depleted in adult females (2-month-old)[39]. By contrast, we observe a complete rescue of ovary weight in *Sws1$^{-/-}$ Chk2$^{-/-}$* and *Swsap1$^{-/-}$ Chk2$^{-/-}$* adult mice and follicles at different stages of development, unlike ovaries from either *Sws1$^{-/-}$* or *Swsap1$^{-/-}$* mice (Fig. 4a, b and Supplementary Fig. 6a, b). Thus, CHK2 is critical for oocyte elimination in the Shu mutants. The rescue by CHK2 ablation is likely better in the Shu mutants than that observed in *Dmc1$^{-/-}$* mice because DSB repair is more proficient,

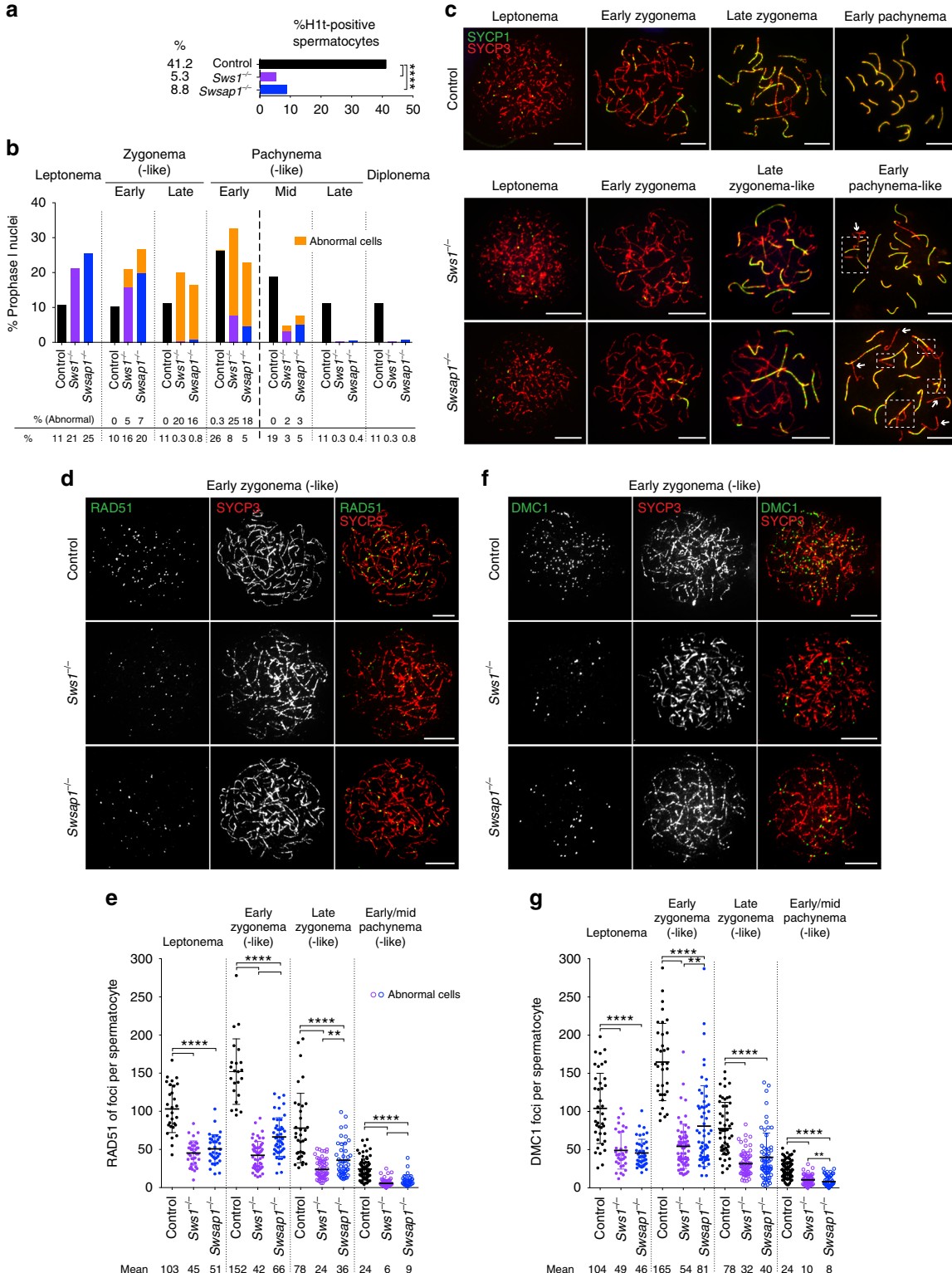

consistent with previous evidence that DSB load is a determinant of oocyte elimination[40].

Importantly, $Sws1^{-/-}$ $Chk2^{-/-}$ and $Swsap1^{-/-}$ $Chk2^{-/-}$ ovaries contain primordial follicles, although still fewer than in controls (Fig. 4c, d and Supplementary Fig. 6b), and, moreover, as observed in $Swsap1^{-/-}$ $Chk2^{-/-}$ mice, synapsis defects persist (open circles; Fig. 4e), as do reduced MLH1 focus numbers (Fig. 4e and

Supplementary Fig. 6c). Nonetheless, $Sws1^{-/-}$ $Chk2^{-/-}$ and $Swsap1^{-/-}$ $Chk2^{-/-}$ mutant females produce offspring, producing about half the litter size of controls, and the female offspring themselves are fertile ($Swsap1^{-/-}$ $Chk2^{-/-}$, Supplementary Table 3b, c). This rescue is remarkable, given the absence of MLH1 foci on one or more chromosomes in every double mutant oocyte scored ($Swsap1^{-/-}$ $Chk2^{-/-}$, Fig. 4e and Supplementary Fig. 6c).

**Fig. 2** SWS1-SWSAP1 is required for meiotic homolog synapsis and RAD51 and DMC1 focus assembly. **a** Histone H1t staining indicates that few spermatocytes in *Sws1* and *Swsap1* mutants progress to mid-pachynema and beyond. Total number of mid-pachytene, late-pachytene, and diplotene spermatocytes divided by the total number of spermatocytes analyzed from adult testes. Mice: control, $n = 6$; $Sws1^{-/-}$, $Swsap1^{-/-}$, $n = 3$. ****, $P \leq 0.0001$; Fisher's exact test, two-tailed. **b**, **c** *Sws1* and *Swsap1* mutants show altered meiotic progression and abnormal chromosome synapsis. Percentage of spermatocytes in each of the indicated meiosis prophase I stages in **b** with representative images in **c**. $Sws1^{-/-}$ and $Swsap1^{-/-}$ cells at leptonema are indistinguishable from controls. At later stages abnormal cells with synaptic defects are observed; because synapsis is abnormal these stages are appended with the word "-like" (orange bars). At early zygonema, chromosomes begin to synapse in the majority of mutant cells, although the number of synaptic stretches are usually reduced; however, delayed synapsis onset is apparent in a subset of mutant cells, as indicated by the lack of SYCP1 stretches. At late zygonema, abnormal cells with fully formed chromosome axes (SYCP3) but little or no synapsis (SYCP1) predominate. Fully-synapsed chromosomes with thicker and shorter SYCP3 axes characterize early pachynema, indicative of chromatin condensation, however, abnormal cells with unsynapsed chromosomes and chromosomes with non-homologous synapsis and/or partial asynapsis are frequent in the mutants. Mid-pachytene (H1t-positive) abnormal cells contain unsynapsed chromosomes, broken bivalents, or parts of chromosomes involved in non-homologous synapsis. Scale bars, 10 μm. Boxes in **c** in early pachytene-like cells highlight non-homologous synapsis and arrows indicate unsynapsed chromosomes. **d–g** RAD51 and DMC1 focus counts are reduced in *Sws1* and *Swsap1* mutant spermatocytes. Representative chromosome spreads from adult mice at early zygonema (-like) are shown in **d**, **f** with focus counts for all stages in **e**, **g**. $n = 3$. Error bars, mean ± s.d. Scale bars, 10 μm. Each circle in **e**, **g** indicates the total number of foci from a single nucleus. Solid circles in **e**, **g** normal cells. Open circles in **e**, **g** cells with abnormal synapsis. Abnormal cells at early zygonema in the *Sws1* and *Swsap1* mutants are not indicated because they are a fraction of cells and are not distinguishable without SYCP1 costaining. **$P \leq 0.01$; ****$P \leq 0.0001$; Mann–Whitney test, one-tailed

To address the basis for the smaller litter size in the $Swsap1^{-/-}$ $Chk2^{-/-}$ double mutants, corpora lutea from pregnant females at 12.5 dpc (or in a few cases from adult virgin mice) were counted in the ovary as a measure of ovulated oocytes. We also counted the total number of implantation sites in the uterine horns, distinguishing those with normal versus resorbed embryos. Both the number of ovulated oocytes and the number of implantation sites are marginally reduced (~20%) in $Swsap1^{-/-}$ $Chk2^{-/-}$ females, although neither reaches statistical significance (Supplementary Fig. 6d). The number of embryos is reduced (38%) and concomitantly, the number of resorbed embryos increased (3-fold), both of which reach statistical significance. These results suggest that the embryo lethality is increased, possibly due to aneuploidy caused by chromosome missegregation from reduced crossovers, but a reduced number of ovulated oocytes may also contribute to the smaller litters.

Unlike females, $Sws1^{-/-}$ $Chk2^{-/-}$ and $Swsap1^{-/-}$ $Chk2^{-/-}$ males exhibit only minimal rescue. While testes are not significantly larger (Fig. 4a and Supplementary Fig. 6e), some tubules exhibit improved cellularity and round and elongated spermatids are occasionally observed (Fig. 4a, g and Supplementary Fig. 6f). H1t-positive spermatocytes are also increased in number ($Swsap1^{-/-}$ $Chk2^{-/-}$, Fig. 4f). Nonetheless, tubules still mostly lack the full complement of germ cells (Fig. 4g and Supplementary Fig. 6f) and mice remain infertile (Supplementary Table 3b). The minimal rescue by CHK2 loss in males could reflect a DSB-independent arrest tied to sex-body defects[36].

**BRCA2 C terminus promotes *Swsap1* spermatocyte progression.** Because some chromosomes in Shu-mutant meiocytes synapse and some cells progress to have MLH1 foci, we reasoned that another mediator protein(s) promotes some level of DSB repair in the absence of the Shu complex. One obvious candidate is BRCA2. Loss of BRCA2 in testis abolishes assembly of both RAD51 and DMC1 into foci, such that spermatocytes do not progress past early pachynema[9]. Although expression of BRCA2 lacking the C-terminal domain in $Brca2^{\Delta27/\Delta27}$ mice[41] causes an HR defect in somatic cells[42], there is little discernible effect on RAD51 and DMC1 foci in early meiotic cells and cells progress to later prophase I stages (Fig. 5a–g). These mice express a truncated BRCA2 protein that lacks C-terminal RAD51 and DMC1 interaction sites[5–7], but contains several other RAD51 and DMC1 interaction sites which can promote their assembly into foci[2,8] (Fig. 5d–g). $Brca2^{\Delta27/\Delta27}$ mice are fertile, although testes are

smaller (Supplementary Fig. 7a), possibly due to late meiotic prophase defects.

We asked whether the BRCA2 C terminus plays a role in the absence of an intact Shu complex to support RAD51 and DMC1 foci and thus inter-homolog repair. Indeed, loss of the BRCA2 C terminus aggravates the homolog synapsis defects in $Swsap1^{-/-}$ mice, and H1t-positive cells are absent from $Swsap1^{-/-}$ $Brca2^{\Delta27/\Delta27}$ mice, indicating a fully penetrant meiotic arrest (Fig. 5a–c). Testis weights are also slightly reduced (Supplementary Fig. 7a). Unlike $Swsap1^{-/-}$ spermatocytes, most double mutant cells at early zygonema show delayed synapsis, and all cells at early pachynema display asynapsis and/or nonhomologous synapsis, with fewer fully synapsed chromosomes (Fig. 5b, c and Supplementary Fig. 7b). Importantly, $Swsap1^{-/-}$ $Brca2^{\Delta27/\Delta27}$ spermatocytes show even fewer foci at leptonema and early zygonema for both RAD51 (2.4 and 1.6-fold reduction, respectively) and DMC1 (1.1 and 1.6-fold reduction, respectively) compared with $Swsap1^{-/-}$ (Fig. 5d–g). There is a concomitant further increase in MEIOB foci (1.2-fold; Fig. 5h), although interestingly, MEIOB foci are still fewer than in $Dmc1^{-/-}$ spermatocytes. We conclude that, although the BRCA2 C terminus is largely dispensable for meiotic HR in the presence of SWSAP1, it functions in the absence of an intact Shu complex to support stable assembly of RAD51 and DMC1 on resected DNA ends. However, it is clearly not sufficient to overcome the loss of the Shu complex at most DSBs.

## Discussion

While canonical RAD51 paralogs are required for embryonic development in mice[2], we found that the non-canonical RAD51 paralog SWSAP1 and its interacting partner SWS1 are not essential for mouse viability. However, both proteins are required for normal meiotic progression. We further show that the meiotic phenotypes of double mutant *Sws1 Swsap1* mice are similar to the single mutants, indicating that they are epistatic to each other and function together. *Sws1* and *Swsap1* mutants demonstrate ~3-fold reduction in RAD51 and DMC1 focus formation in meiosis that ultimately results in DSB repair and homolog synapsis defects that are sufficient to render the mice infertile. Thus, our studies reveal a crucial requirement for the mouse Shu complex in meiosis, unlike Shu complexes in other organisms in which meiotic progression is only partially affected or not affected at all.

The embryonic lethality associated with mutation of canonical RAD51 paralogs makes it unclear how critical these proteins are for meiotic progression. One hint comes from mice with a

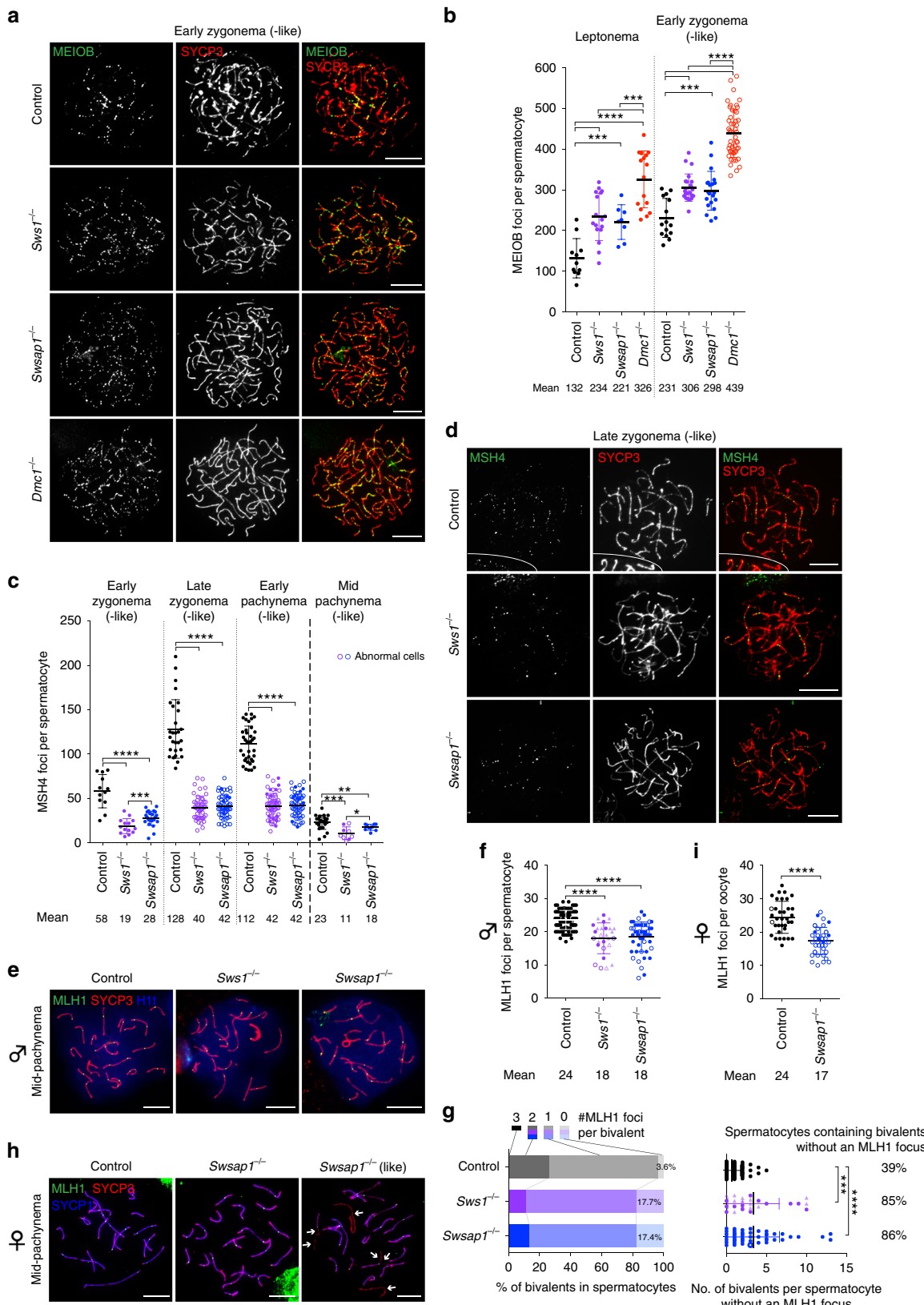

hypomorphic *Rad51c* allele, a fraction of which are infertile due to a substantially reduced level of RAD51C expression[43]. Spermatocytes from infertile mice show decreased RAD51 focus formation, although DMC1 was not examined. Thus, as in mitotic cells, multiple protein complexes are likely needed to promote recombinase activity in meiotic cells, although how they functionally interact remains to be elucidated. Since the Shu knockout mice are viable, they have provided a unique opportunity to determine the contribution of a RAD51 paralog to meiotic HR.

**Fig. 3** *Sws1* and *Swsap1* mutants accumulate resected DNA and have reduced HR foci later in meiosis. **a, b** MEIOB focus counts are increased in *Sws1*$^{-/-}$ and *Swsap1*$^{-/-}$ spermatocytes, but not as much as in *Dmc1*$^{-/-}$ spermatocytes. Representative chromosome spreads at early zygonema (-like) from adult mice in **a** with focus counts at indicated stages in **b**. Mice: Control, $n = 2$; *Sws1*$^{-/-}$, *Swsap1*$^{-/-}$, $n = 3$, *Dmc1*$^{-/-}$, $n = 4$. **c, d** MSH4 focus counts are reduced in *Sws1*$^{-/-}$ and *Swsap1*$^{-/-}$ spermatocytes. Focus counts at indicated stages in **c** with representative images of chromosome spreads at late zygonema (-like) in **d**. White lines in the images demark boundaries of other cells in the field. Mice: control, $n = 2$; *Sws1*$^{-/-}$, *Swsap1*$^{-/-}$, $n = 3$. **e, f** MLH1 focus counts are reduced on average by ~20% in *Sws1*$^{-/-}$ and *Swsap1*$^{-/-}$ spermatocytes at mid-/late pachynema (-like). Representative chromosome spreads at mid-pachynema in **e** with focus counts in **f**. Mice: Control, $n = 4$; *Swsap1*$^{-/-}$ (Δ131, circles; (+1), triangles), *Sws1*$^{-/-}$, $n = 3$. **g** Although most *Sws1*$^{-/-}$ and *Swsap1*$^{-/-}$ bivalents have at least one MLH1 focus (left), most spermatocytes have at least one bivalent which lacks a focus. Left, percentage of bivalents in mid-/late pachynema (-like) with indicated number of MLH1 foci per bivalent. Right, number of bivalents without an MLH1 focus per spermatocyte; total percentage of spermatocytes containing bivalents without an MLH1 focus is indicated. Solid circles and triangles, spermatocytes with fully synapsed homologs. Open circles, abnormal spermatocytes containing homologs partially synapsed and/or one or two homolog pairs fully unsynapsed. **h, i** MLH1 focus counts are reduced on average by ~30% in *Swsap1*$^{-/-}$ oocytes. Representative chromosome spreads at mid-pachynema (-like) from mice at embryonic day 18.5 in **h** with focus counts in **i**. Control, $n = 2$; *Swsap1*$^{-/-}$, $n = 3$. While a fraction of mutant oocytes show completely synapsed bivalents (solid circles in **i**), the majority have chromosomal synapsis defects (open circles), i.e., fully unsynapsed and/or partially synapsed bivalents. In controls, a small fraction of oocytes also have one or two chromosome ends that are not fully synapsed. Each circle in **b, c, f, i** indicates the total number of foci from a single nucleus. Solid circles, normal cells. Open circles/triangles, cells with abnormal synapsis. Scale bars in **a, d, e, h**, 10 μm. Error bars in **b, c, f, g, i**, mean ± s.d. *$P \leq 0.05$; **$P \leq 0.01$; ***$P \leq 0.001$; ****$P \leq 0.0001$; Mann–Whitney test, one-tailed

---

Formally, the reduction in RAD51 and DMC1 focus formation in Shu mutant spermatocytes could be due to decreased DSB formation during meiosis or impaired resection of DNA ends. However, we find that γH2AX levels at early prophase I stages are similar to controls and foci of the ssDNA-binding protein MEIOB are increased in the *Sws1* and *Swsap1* mutants, implying that resected DNA ends accumulate. The number of MEIOB foci are not as sharply increased as when DMC1 itself is absent, consistent with the ability of a fraction of RAD51 and DMC1 foci to form in the absence of the Shu complex.

These results position the mammalian Shu complex as a critical mediator protein for both RAD51 and DMC1 nucleoprotein filament formation/stabilization. It is interesting to note that budding yeast Rad51 has an indirect role in promoting DNA strand exchange in meiosis by acting as an accessory factor for Dmc1[44]. Although not known in the mouse, RAD51 appears to play a similarly indirect role in Arabidopsis[45]. Thus, it remains possible that the mouse Shu complex acts indirectly on DMC1 through RAD51. However, unlike in mouse, the Shu complex in budding yeast, sometimes termed PCSS, only affects Rad51 focus formation; Dmc1 focus formation is unaffected, suggesting that any affect the Shu complex has on Rad51 is not transmitted to Dmc1[15]. Thus, we believe it is more likely that the mouse Shu complex is directly supporting DMC1 function rather than acting through RAD51.

We envision the mouse Shu complex stabilizing both RAD51 and DMC1 nucleoprotein filaments (Fig. 5i), and possibly remodeling them, as reported for RAD51 by Shu complexes and canonical RAD51 paralogs in other organisms[15,18,46]. By contrast, the primary meiotic role of full-length BRCA2 is nucleation of RAD51 and DMC1 nucleoprotein filaments[9], a role ascribed to the BRC repeats in the center of the protein[2,8]. Notably, the BRCA2 C terminus has also been implicated in RAD51 filament stabilization by selectively binding to the interface between two RAD51 protomers[5,6]. If so, however, our data suggests that this stabilization is dispensable in meiotic recombination unless the Shu complex is disrupted, in which case meiotic progression is completely abolished. It will be interesting to determine if the BRCA2 C terminus also promotes stabilization of DMC1 filaments as it does for RAD51 filaments. Collectively, our data suggests that while the Shu complex and the BRCA2 C terminus have overlapping biochemical roles during meiotic HR, the Shu complex is more critical given the infertility of Shu mutant mice.

About a fifth of Shu mutant spermatocytes proceed to mid-pachynema to form crossovers marked by MLH1 foci. Similar to the reduction in RAD51 and DMC1 foci at early zygonema, Shu mutant spermatocytes show a substantial decrease in MSH4 foci at early pachynema, with none of the spermatocytes having MSH4 focus numbers within one standard deviation of the mean seen in control cells (Fig. 3c). MSH4 is critical for the stabilization of strand exchange intermediates, a fraction of which resolve as crossovers. Considering that the average reduction in the number of MSH4 foci is ~3-fold at early pachynema, yet the reduction in MLH1 foci is only ~20%, these results imply that mutant cells are tightly regulating the number of crossovers despite the severe reduction in recombination intermediates. Thus, this study demonstrates a clear example of crossover homeostasis[47,48] operating in a mouse DSB repair mutant.

We found that loss of the DNA damage checkpoint kinase CHK2 restores fertility to *Sws1* and *Swsap1* mutant females, despite the persistence of synaptic defects and reduced MLH1 foci. Fertility of females with a hypomorphic *Trip13* mutation has previously been restored by deactivating the oocyte DNA damage checkpoint through CHK2 or p53/TAp63 ablation[39], although in this mutant MLH1 focus numbers in oocytes are inferred to be nearly normal (as reported for spermatocytes[49,50]). It would be interesting to determine whether viable pups from *Sws1*$^{-/-}$ *Chk2*$^{-/-}$ and *Swsap1*$^{-/-}$ *Chk2*$^{-/-}$ dams arise from oocytes with an MLH1 focus on almost every chromosome (Supplementary Fig. 6c) or from fortuitous segregation of a number of non-recombinant chromosomes[51].

In summary, our work provides a comprehensive analysis of meiotic phenotypes of *Sws1* and *Swsap1* knockout mice which clearly could not have been anticipated from published work on the Shu complexes in other organisms, or from the work on other mammalian RAD51 paralogs. Moreover, discovering a mutant that has reduced, but not completely eliminated recombination, has allowed us to uncover robust crossover homeostasis, fertility despite crossover defects, and a functional interaction with BRCA2.

## Methods

**Mouse care.** The care and use of mice in this study were performed in accordance with a protocol approved by the Institutional Animal Care and Use Committee (IACUC) at Memorial Sloan Kettering Cancer Center (MSKCC). Mice were housed under Federal regulations and policies governed by the Animal Welfare Act (AWA) and the Health Research Extension Act of 1985 in the Research Animal Resource Center (RARC) at MSKCC, and was overseen by IACUC.

**Generation and genotyping of Shu mutant mice.** We targeted *Sws1* and *Swsap1* with TALE nucleases directed to each gene's first exon, close to the translational start sites. TALEN pairs (RNA) were injected into fertilized mouse eggs, derived from superovulated CBA/J x C57BL/6 J F1 females mated with C57BL/6 J males,

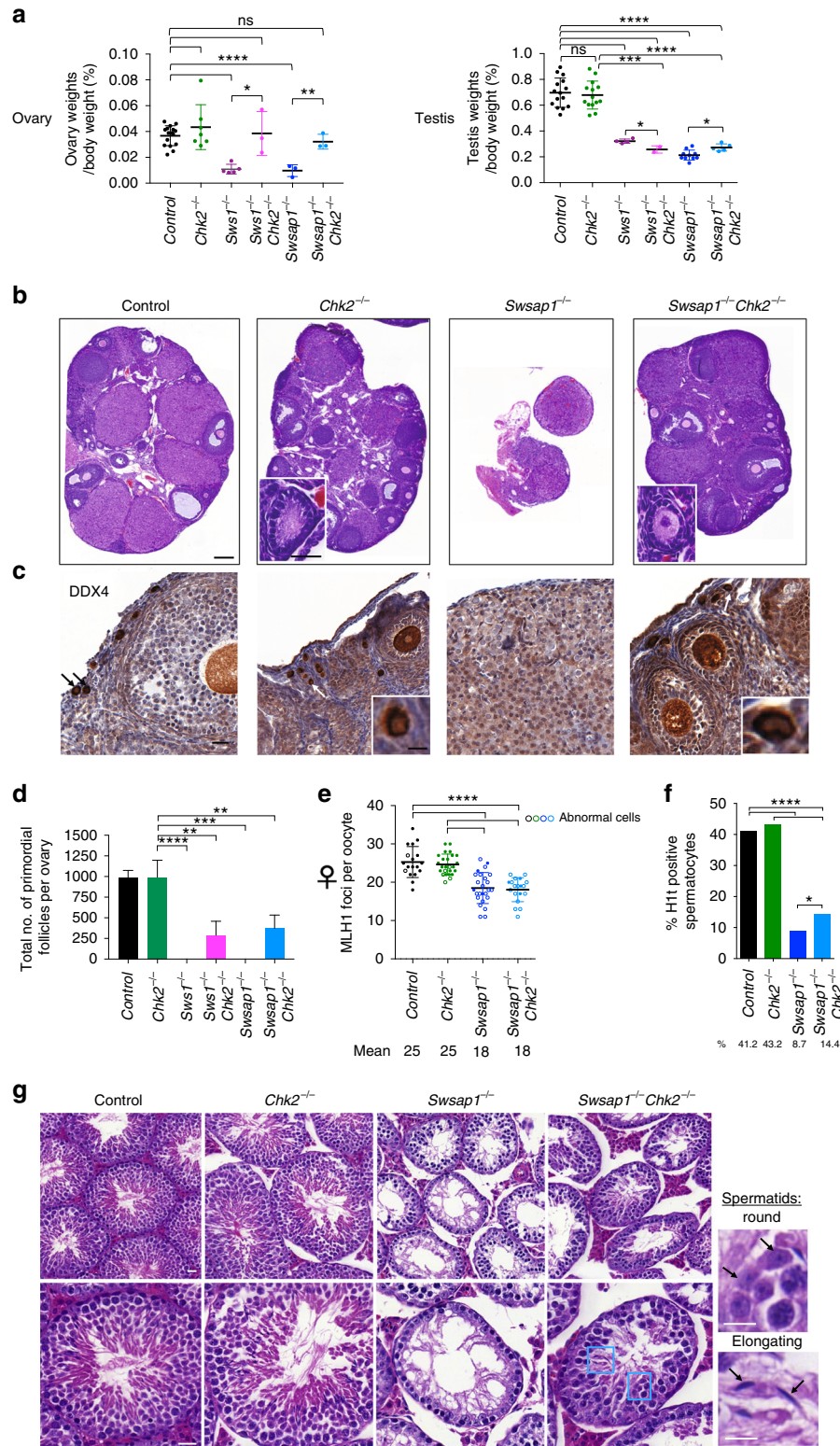

which were then implanted into pseudo-pregnant females[52,53]. To initially genotype founder mice, at least 10 cloned PCR products from each of 22 *Sws1* and 10 *Swsap1* founders were sequenced. Founders were backcrossed to C57BL/6 J to separate multiple alleles and then further backcrossed for 3–6 additional generations prior to generating experimental mice.

Genotyping for *Sws1* was done by PCR-sequencing using the following PCR primers: *Sws1*-A: 5′-CCTGCAGGGCGCGTGAAGTTC-3′, *Sws1*-B: 5′-ACCGGCT CGCACTCAGGGATC-3′ under the following conditions: 94 °C, 3 min; 35 cycles of 94 °C, 30 s; 55 °C, 1 min and 65 °C, 30 s; and a final extension of 72 °C, 5 min.

The PCR product (259 bp) was sequenced using *Sws1*-A primer and sequencing reads were aligned against the wild-type controls to detect the 1-bp deletion.

Genotyping for *Swsap1* was done using the following PCR primers *Swsap1*-C: 5′-TCTGTGAACTATAGCCAATGAGGC-3′, and *Swsap1*-D: 5′-AACTGTCAC TCAGGCGCGAACTAG-3′ under the following PCR conditions: 94 °C, 3 min; 35 cycles of 94 °C, 30 s; 55 °C, 1 min and 65 °C, 30 s; and a final extension of 72 °C, 5 min. The *Swsap1(+1)* allele was genotyped by PCR-sequencing using the *Swsap1*-C primer; the *Swsap1Δ131* allele was genotyped by running PCR products on a 2.4% agarose gel. The wild-type product is 396 bp and the mutant is 265 bp.

**Fig. 4** *Chk2* mutation rescues fertility of *Sws1* and *Swsap1* mutant females. **a** Ovary to body weight ratios of *Sws1⁻/⁻ Chk2⁻/⁻* and *Swsap1⁻/⁻ Chk2⁻/⁻* adult mice are similar to controls, but testis to body weight ratios remain significantly reduced. Female mice: Control, n = 16; *Chk2⁻/⁻*, n = 7; *Sws1⁻/⁻*, n = 5; *Sws1⁻/⁻ Chk2⁻/⁻*, n = 3; *Swsap1⁻/⁻*, *Swsap1⁻/⁻ Chk2⁻/⁻*, n = 3. Male mice: control, n = 15; *Chk2⁻/⁻*, n = 14; *Sws1⁻/⁻*, n = 4; *Sws1⁻/⁻ Chk2⁻/⁻*, n = 2; *Swsap1⁻/⁻*, n = 10; *Swsap1⁻/⁻ Chk2⁻/⁻*, n = 5. ns not significant; *P ≤ 0.05; **P ≤ 0.01; ***P ≤ 0.001; Student's *t*-test; two-tailed. **b–d** Adult *Swsap1⁻/⁻ Chk2⁻/⁻* ovaries have follicles at various stages of oocyte development, unlike *Swsap1⁻/⁻* ovaries. Primordial follicles in *Chk2⁻/⁻* and *Swsap1⁻/⁻ Chk2⁻/⁻* are highlighted in **b** in insets and by arrows in **c**; primordial follicles in *Chk2⁻/⁻*, *Sws1⁻/⁻ Chk2⁻/⁻* and *Swsap1⁻/⁻ Chk2⁻/⁻* are quantified in **d**. Sections were stained with hematoxylin and eosin (H&E) in **b**, and with hematoxylin and an antibody for DDX4/Vasa in **c**. Scale bar, 500 μm and 50 μm insets. Mice: control, n = 4; *Chk2⁻/⁻*, n = 5; *Sws1⁻/⁻*, n = 4; *Sws1⁻/⁻ Chk2⁻/⁻*, n = 3; *Swsap1⁻/⁻*, *Swsap1⁻/⁻ Chk2⁻/⁻*, n = 3. **P ≤ 0.01; ***P ≤ 0.001; ****P ≤ 0.0001; Student's *t*-test, two-tailed. **e** MLH1 foci are reduced in *Swsap1⁻/⁻ Chk2⁻/⁻* oocytes from mice at embryonic day 18.5 similar to the level observed in *Swsap1⁻/⁻* oocytes, and synaptic defects are also observed in the majority of *Swsap1⁻/⁻ Chk2⁻/⁻* oocytes as also seen in *Swsap1⁻/⁻* mice (open circles). Mice: Control, *Swsap1⁻/⁻*, n = 1 (from Fig. 3i); *Swsap1⁻/⁻ Chk2⁻/⁻*, *Chk2⁻/⁻*, n = 1. ****P ≤ 0.0001; Mann–Whitney test, one-tailed. **f** *Swsap1⁻/⁻ Chk2⁻/⁻* spermatocytes that progress to mid-pachynema and beyond are increased in number relative to that observed in *Swsap1⁻/⁻* mice, as demonstrated by histone H1t staining. Total number of mid-pachytene, late-pachytene, and diplotene spermatocytes divided by the total number of spermatocytes analyzed from adult testes. Mice: Control, *Swsap1⁻/⁻*, from Fig. 2a; *Chk2⁻/⁻*, *Swsap1⁻/⁻ Chk2⁻/⁻*, n = 2. *P ≤ 0.05; ****P ≤ 0.0001; Fisher's exact test, two-tailed. **g** Spermatocytes in adult *Swsap1⁻/⁻ Chk2⁻/⁻* mice mostly arrest at pachynema, but tubules occasionally contain round and elongated spermatids (blue rectangles and insets). Sections were stained with H&E. Scale bars, 100 μm (top panel), 50 μm (bottom panel), and 20 μm (inset). Error bars in **a**, **d**, **e**, mean ± s.d

*Chk2* (refs. [54,55]) and *Brca2^Δ27* (refs. [41,42]) mice and genotyping were previously described.

**RT-PCR**. Twenty milligrams of mouse tissue was incubated with 1 ml Triazol and homogenized with a Dounce homogenizer. The extract was transferred to Eppendorf tubes and incubated for 5 min at room temperature. Extracts were centrifuged at 12,000×g for 10 min at 4 °C. Supernatants were transferred to another Eppendorf tube and RNA was extracted using chloroform followed by isopropanol precipitation. The RNA pellet was dissolved in $H_2O$. To prepare the cDNA library, Superscript one-step RT-PCR kit was used (Invitrogen). To amplify cDNA for *Sws1* and *Swsap1*, the following primers were used: *Sws1*-RT-A: 5′-AA GTTCGCAGCGCCCGGG-3′, *Sws1*-RT-B: 5′-CTAGGCTTCTGTCTTTGAAG TCC-3′, *Swsap1*-RT-A: 5′-ATGGCGGAGGCGCTGAGG-3′, *Swsap1*-RT-B: 5′-TC AGGTCTTTGAATCTGCACCTG-3′. The following conditions were used for PCR: 94 °C, 2 min; 30 cycles of 94 °C, 1 min; 65 °C, 1 min and 72 °C, 1 min; and a final extension of 72 °C, 10 min. PCR products were separated on 1.0% agarose gels, excised, and DNA was purified and sequenced. The *Sws1*-RT-A and *Swsap1*-RT-A primers were used for sequencing.

### Primer sequences

| Primer name | 5′ to 3′ sequence | Usage |
| --- | --- | --- |
| *Sws1*-A | CCTGCAGGGCGCGTGAAGTTC | Genotyping for *Sws1* |
| *Sws1*-B | ACCGGCTCGCACTCAGGGATC | |
| *Swsap1*-C | TCTGTGAACTATAGCCAATGAGGC | Genotyping for *Swsap1* |
| *Swsap1*-D | AACTGTCACTCAGGCGCGAACTAG | |
| *Sws1*-RT-A | AAGTTCGCAGCGCCCGGG | To amplify cDNA for *Sws1* |
| *Sws1*-RT-B | CTAGGCTTCTGTCTTTGAAGTCC | |
| *Swsap1*-RT-A | ATGGCGGAGGCGCTGAGG | To amplify cDNA for *Swsap1* |
| *Swsap1*-RT-B | TCAGGTCTTTGAATCTGCACCTG | |

**Histology**. Ovaries and testes were dissected from animals at the stated ages and (1) fixed in Bouin's and stained with PAS and counterstained with hematoxylin, (2) fixed in 4% PFA and stained with H&E, or (3) fixed in 4% PFA and stained with hematoxylin and antibodies against DDX4/Vasa (Abcam, ab13840; 2.5 μg/ml) or c-Kit (Cell Signaling, 3074; 0.75 μg/ml) or were TUNEL-stained (Roche, 03333566001 and 11093070910). Staging of PAS- or H&E-stained testes sections was performed as described[25]. For follicle counts, ovaries were serially sectioned at 6 μm thickness, and follicles were counted in every fifth section, without further correction. The results were from one ovary from each animal. To determine the number of implantation sites, pregnant females at 12.5 dpc were dissected and the number of implantation sites were counted in both uterine horns and distinguished based on the presence of a normal embryo or resorbed embryo. For ovulated oocytes, ovaries were collected from pregnant females at 12.5 dpc, fixed in 4% PFA, sectioned in the center, stained with H&E, and the number of corpora lutea were counted. Reported results are the total from both ovaries from each animal.

**Testis extracts and western blotting**. To prepare testis extracts, adult mouse testes were macerated in NETN buffer (100 mM NaCl, 20 mM Tris-HCl pH 8.0, 0.5 mM EDTA, 0.5% (v/v) Nonidet P-40 (NP-40)) with protease inhibitor cocktail (Roche) and incubated for 30 min at 4 °C with gentle flicking every few minutes to resuspend the extracts in the buffer. The supernatant was collected after spinning the samples at 20,000×g for 10 min at 4 °C. Thirty micrograms of protein were loaded on a precast SDS PAGE gel (Bio-Rad), transferred onto nitrocellulose membrane (162-0145, Bio-Rad), and blocked with 5% milk in PBST (50 mM Tris-HCl pH 7.5, 150 mM NaCl, 0.05% Tween 20) for 1 h. For immunodetection, the following antibodies were used: rabbit anti-RAD51 (Calbiochem, PC130; 1:5000), mouse anti-DMC1 (Santa Cruz Biotechnology, sc-373862; 1:500) and mouse anti-tubulin (Sigma, T9026; 1:10,000). Secondary antibodies used were peroxidase-linked anti-mouse or anti-rabbit IgG (1:10,000; GE Healthcare).

**Spermatocyte chromosome spreads and immunofluorescence**. Testes were collected from 2–4-month-old mice and spermatocytes were prepared for surface spreading and processed using established methods for immunofluorescence[56], using the following primary antibodies in dilution buffer (0.2% BSA, 0.2% fish gelatin, 0.05% Triton X-100, 1xPBS), with incubation overnight at 4 °C: mouse anti-SYCP3 (Santa Cruz Biotechnology, sc-74569; 1:200), rabbit anti-SYCP3 (Abcam, ab15093; 1:500), goat anti-SYCP3 (Santa Cruz Biotechnology, sc-20845; 1:200), rabbit anti-SYCP1 (Novus, NB-300-229; 1:200), mouse anti-γH2AX (Millipore, 05-636; 1:500), rabbit anti-RAD51 (Calbiochem, PC130; 1:200), rabbit anti-DMC1 (Santa Cruz Biotechnology, sc-22768; 1:200), rabbit anti-MEIOB (kindly provided by P.J. Wang, University of Pennsylvania; 1:200), rat anti-RPA2 (Cell Signaling Technology, 2208 s; 1:100), rabbit anti-MSH4 (Abcam, ab58666; 1:100), mouse anti-MLH1 (BD Biosciences, 51-1327GR; 1:50) and guinea pig anti-H1t (kindly provided by M.A. Handel, Jackson Laboratory; 1:500). Slides were subsequently incubated with the following secondary antibodies at 1:200 to 1:500 dilution for 1 h at 37 °C: 488 donkey anti-mouse (Life Technologies, A21202), 488 donkey anti-rabbit (Life Technologies, A21206), 488 goat anti-rat (Life Technologies, A11006), A568 goat anti-mouse (Molecular probes, A-11019), 568 goat anti-rabbit (Life Technologies, A11011), 594 donkey anti-mouse (Invitrogen, A21203), 594 goat anti-rabbit (Invitrogen, A11012), donkey 594 anti-goat (Invitrogen, A11058), 647 donkey anti-mouse (Life Technologies, A31571), 647 donkey anti-rabbit (Invitrogen, A31573), and 647 goat anti-guinea pig (Life Technologies, A21450). Cover slips were mounted with ProLong Gold antifade reagent with or without DAPI (Invitrogen, P36935 and P36934, respectively). Immunolabeled chromosome spread nuclei were imaged on a Marianas Workstation (Intelligent Imaging Innovations; Zeiss Axio Observer inverted epifluorescent microscope with a complementary metal-oxide semiconductor camera) using 100× oil-immersion objective. Images were processed using Image J for foci analysis and Photoshop (Adobe) to make the figures. Spermatocytes were staged by assessing the extent of SYCP3 staining and synapsis (based on SYCP1 staining for Figs. 2, 5 and Supplementary Fig. 3). Only foci colocalizing with the chromosome axis were counted. In controls, leptotene cells are characterized by the presence of small stretches of SYCP3 and no SYCP1 staining. At early zygonema, homolog synapsis initiates, marked by the presence of SYCP1, and longer SYCP3 stretches are visible as chromosome axes continue to elongate. By late zygonema, chromosome axis formation completes at the same time that SYCP1 appears between homologs (>50% of overall synapsis). At pachynema, homologs are fully synapsed (co-localization of SYCP3 and SYCP1), except in the non-pseudoautosomal region of the XY chromosomes. Due to chromatin condensation at this stage, pachytene chromosome axes are shorter and thicker. H1t staining is used whenever possible to distinguish mid-/late-pachytene from early-pachytene cells. Late pachytene cells are further characterized by thickening of chromosome ends and elongation/curling of XY chromosomes. In diplotene cells, chromosome desynapsis ensues. *Sws1⁻/⁻* and *Swsap1⁻/⁻* cells at leptonema are indistinguishable from controls. Abnormal cells in mutants are characterized as follows: Early zygotene-like cells have chromosomes with long axes (SYCP3) and no SYCP1 stretches. Late zygotene-like cells

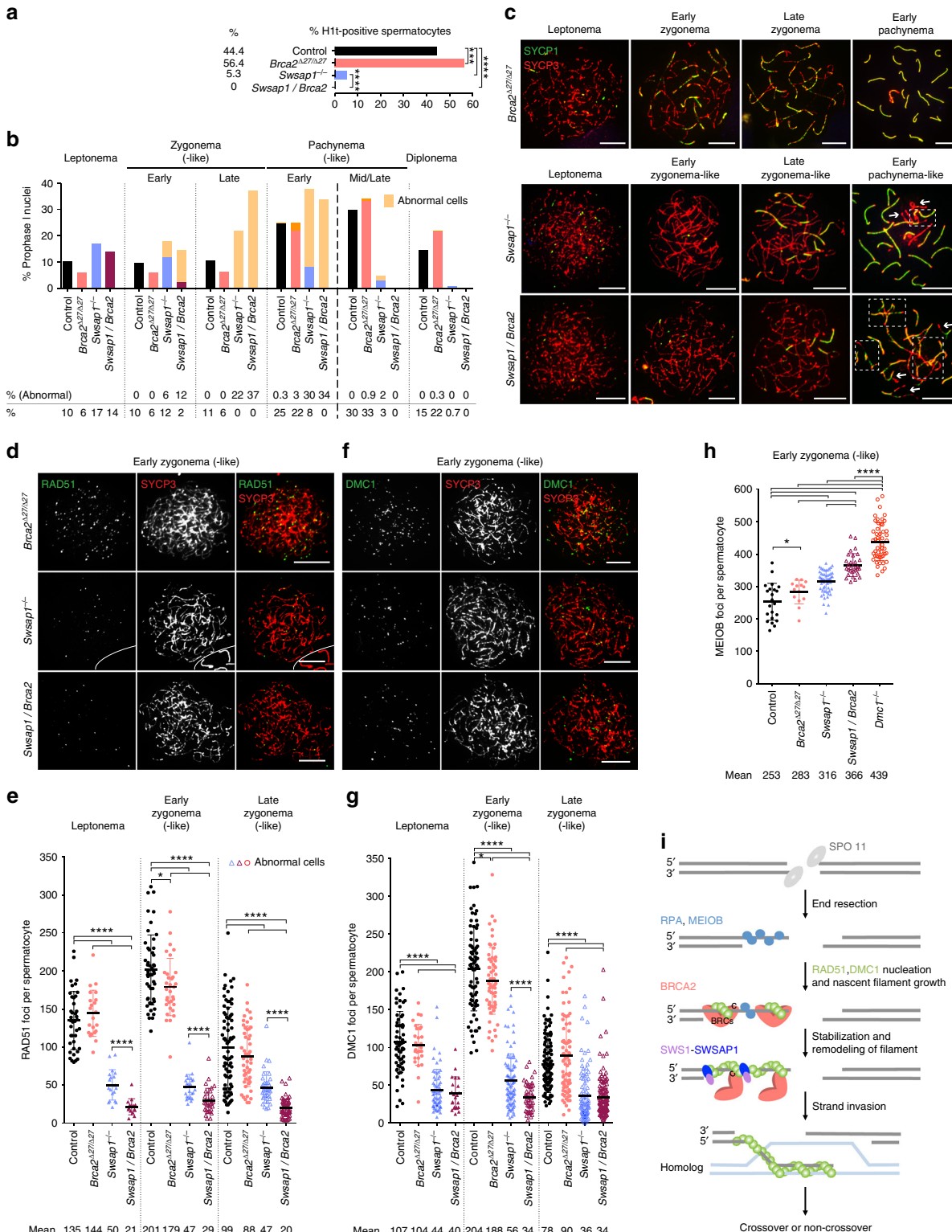

have chromosomes with fully formed axes (SYCP3) and little or no synapsis in addition to elongated chromosomes completing synapsis as in controls. Early pachytene-like cells have unsynapsed chromosomes and/or incompletely synapsed homologs as well as non-homologous synapsis, but also display fully synapsed autosomes with thicker and shorter SYCP3 axes indicative of chromatin condensation characteristic of this stage. Mid-pachytene-like cells, which are H1t-positive, also display synaptic abnormalities and may contain chromosome fragments. Cells displaying normal homolog synapsis but chromosome end-to-end fusions are considered normal cells in control and mutants.

**Oocyte chromosome spreads**. Prenatal ovaries were collected at embryonic day 18.5 and processed to obtain oocyte spreads as described[57] with some modifications. Briefly, ovaries were placed in a 1.5 ml Eppendorf tube containing 0.7 ml isolation medium (TIM: 104 mM NaCl, 45 mM KCl, 1.2 mM MgSO$_4$, 0.6 mM KH$_2$PO$_4$, 0.1% (w/v) glucose, 6 mM sodium lactate, 1 mM sodium pyruvate, pH 7.3, filter sterilized) and fragmented by pipetting up and down several times. After centrifuging for 3 min at 400 $g$ and discarding the supernatant, 0.5 ml of 1 mg/ml collagenase (Sigma, C0130) in TIM was added to the ovarian fragments and tubes were incubated at 37 °C for 1 h with gentle shaking. Next, careful pipetting up and

**Fig. 5** BRCA2 C terminus promotes *Swsap1* spermatocyte progression. **a** *Swsap1*$^{-/-}$(+1) *Brca2*$^{\Delta27/\Delta27}$ spermatocytes do not progress past early pachynema in contrast to *Swsap1*$^{-/-}$(+1) in which a fraction progresses. Mice: Control, n = 8 (combined with those from Fig. 2a); *Brca2*$^{\Delta27/\Delta27}$, n = 3; *Swsap1*$^{-/-}$(+1) *Brca2*$^{\Delta27/\Delta27}$, n = 4; *Swsap1*$^{-/-}$(+1), n = 4 (combined with those from Supplementary Fig. 3a). ***P ≤ 0.001; ****P ≤ 0.0001; Fisher's exact test, two-tailed. **b**, **c** *Swsap1*$^{-/-}$(+1) *Brca2*$^{\Delta27/\Delta27}$ spermatocytes show altered meiotic progression and aggravated synapsis defects relative to *Swsap1*$^{-/-}$. Percentage of spermatocytes in the indicated stages in **b** with representative images in **c**. Most *Swsap1*$^{-/-}$(+1) *Brca2*$^{\Delta27/\Delta27}$ early-zygotene (-like) cells have delayed synapsis, while is only seen in some *Swsap1*$^{-/-}$(+1) spermatocytes (light orange bars). By early pachynema, all double-mutant spermatocytes display incomplete homolog synapsis and/or nonhomologous synapsis. Boxes in early pachytene-like cells in **c** highlight non-homologous synapsis, and arrows indicate unsynapsed chromosomes. **d**–**g** RAD51 and DMC1 focus counts are further reduced in *Swsap1*$^{-/-}$(+1) *Brca2*$^{\Delta27/\Delta27}$ spermatocytes relative to *Swsap1*$^{-/-}$(+1). Representative chromosome spreads are shown in **d**, **f** with focus counts for the indicated stages in **e**, **g**. Control, n = 5 (RAD51, combined with 2 mice from Fig. 3e) and n = 5 (DMC1, combined with 2 mice from Fig. 2g); *Swsap1*$^{-/-}$(+1), n = 3 (RAD51, combined with that from Supplementary Fig. 3f) and n = 4 (DMC1, combined with that from Supplementary Fig. 3g); *Brca2*$^{\Delta27/\Delta27}$, n = 4 (RAD51) and n = 6 (DMC1); *Swsap1*$^{-/-}$(+1) *Brca2*$^{\Delta27/\Delta27}$, n = 3 (RAD51) and n = 4 (DMC1). **h** MEIOB focus counts at early zygonate (-like) are further increased in *Swsap1*$^{-/-}$(+1) *Brca2*$^{\Delta27/\Delta27}$ spermatocytes relative to *Swsap1*$^{-/-}$(+1), but still lower than that in *Dmc1*$^{-/-}$. Mice: control, n = 3 (combined with those from Fig. 3b); *Swsap1*$^{-/-}$(+1), n = 3 (combined with that from Supplementary Fig. 5c); *Brca2*$^{\Delta27/\Delta27}$, n = 2; *Swsap1*$^{-/-}$(+1) *Brca2*$^{\Delta27/\Delta27}$, n = 3; *Dmc1*$^{-/-}$, n = 4 (from Fig. 3b). Scale bars in **c**, **d**, **f**: 10 μm. Each symbol in **e**, **g**, **h** is the total number of foci from a single nucleus. Solid symbols in **e**, **g**, **h**: normal cells. Open symbols in **e**, **h**: abnormal cells. Error bars in **e**, **g**, **h**: mean ± s.d. *P ≤ 0.05; ****P ≤ 0.0001; Mann–Whitney test, one-tailed. **i** Functional interaction between the mouse Shu complex and the BRCA2 C terminus during meiotic HR. While SWS1-SWSAP1 is critical for stable RAD51 and DMC1 nucleoprotein filament assembly during meiosis, the BRCA2 C terminus can provide some activity in the absence of this complex

down was performed until large fragments of ovarian tissue were no longer visible. Following centrifugation for 3 min at 400 g, the supernatant was discarded, the pellet was carefully resuspended in 0.5 ml 0.05% trypsin (Sigma, T9935), and tubes were incubated at 37 °C for 5 min with gentle shaking. Then 0.5 ml DMEM containing 10% (v/v) FBS was added, carefully pippeted up and down, and centrifuged for 3 min at 400 g. The supernatant was discarded followed by the addition of 1 ml TIM to resuspend the cells by pipetting. Cells were again centrifuged for 3 min at 400 g, and after discarding the supernatant, cells were resuspended in 0.5 ml hypotonic solution (30 mM Tris-HCl pH 8.2, 50 mM sucrose, 17 mM Na-Citrate, 5 mM EDTA, 1× protease inhibitors), and incubated for 30 min to 1 h at room temperature. Positively charged, precleaned glass slides were placed in a humid chamber and a circle of ~1.0–1.5 cm diameter was marked in the center of the slide with a hydrophobic barrier pen; 40 μl of 1% (w/v) PFA containing 0.1% (v/v) Triton X-100 was placed into each circle and 10 μl cell suspension was slowly dropped on each slide within the PFA solution. Slides were slowly dried in a moist chamber that was closed for 2 h, then ajar for 30 min, and then open for 30 min. Slides were rinsed two times with milliQ H$_2$O, and one time with 1:250 Photo-Flo 200 (Kodak, 1464510) solution. Slides were air-dried and stored at -80 °C. Staining of slides was then performed as for the spermatocyte chromosome spreads described above.

**Statistical analysis**. Statistical analyses were performed using a Chi square test for animal breeding, a nonparametric two-tailed Mann–Whitney test for pup analysis, a two-tailed Student's *t*-test for primordial follicles counts in ovaries and testis, ovary, and body weight comparisons, a two-tailed Fisher's test for H1t cell analysis, a nonparametric one-tailed Mann–Whitney test for foci number comparisons, as a normal distribution could not be assumed. Error bars, mean ± s.d.; ns not significant; *P ≤ 0.05; **P ≤ 0.01; ***P ≤ 0.001; ****P ≤ 0.0001.

## Data availability

All relevant data are included in the Supplementary Data File or are available from the authors upon reasonable request.

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

## Acknowledgements

We thank Mary Ann Handel and P. Jeremy Wang for antibodies, John Schimenti for *Dmc1* mice, Kara Bernstein (University of Pittsburgh) and members of the Jasin and Keeney labs for discussions and critical reading of the manuscript, and Katia Manova and members of the MSKCC Molecular Cytology core facility for technical help. This work was supported by MSK Cancer Center Support Grant/Core Grant (NIH P30CA008748), NIH F32GM110978 (R.P.), BFU2016-80370-P (I.R.), R35 GM118092 (S.K.), R35 GM118175 (M.J.), and R01CA185660 (M.J.).

## Author contributions

C.M.A., R.P., P.J.R., S.K., and M.J. designed experiments. C.M.A., R.P., and P.J.R. performed experiments. I.R., S.K., and M.J. supervised the research. C.M.A., R.P., S.K., and M.J. wrote the paper with input from I.R.

## Additional information

**Competing interests:** The authors declare no competing interests.

