## [Peer Review File · Nature Communications]

Reviewers' comments:

Reviewer #1 (Remarks to the Author):

Review of Abreu et al.

The authors explore the roles of 2 Shu complex proteins in mammals by generating mouse mutants. It turns out that the only effects are in meiosis. The interest here is that there is a unique phenotypic consequence of mutating two component genes: impaired (but not abolished) loading of RAD51 and DMC1 recombinases onto resected meiotic DSBs. Although the Shu complex has been investigated in other organisms, the roles in mice have distinctions.

In general, the work is extensive and well presented. The story will be of interest, not only because the role of these proteins is elucidated, but also because the unique phenotypes may be useful for other investigations, for example in checkpoint control (as they did here), and the role of sub-optimal genomic integrity in reproduction and embryonic viability (which they developed to a lesser extent). The overall impact is viewed to be above average, as is the quality of this manuscript.

Major issues:

- a number of data points are derived from only 1 mouse, for example *Sws1*^{-/-} in Fig 3e,f; *Dmc1*^{-/-} in Fig 3a,b; *Sws1*^{-/-} and *Swsap1*^{-/-} oocytes in Fig. 3h,i for *Mlh1* focus counts. The latter is bad, given the relatively small difference in average foci, and the fact that the mice were not fully inbred ("backcrossed for 3-6 additional generations"); there are known to be genetic differences in *Mlh1* foci in different inbred strains. This shortcoming cannot be overcome by simply scoring more spreads from a single mouse.

- the basis for the smaller litter size in *Chk2*^{-/-} *Swsap1*^{-/-} animals should be explored. Mice with smaller oocyte pools may ovulate identical numbers (at least in early life), but may suffer from aneuploidy (associated with reduced crossovers), leading to embryonic lethality. The experiment would be relatively easy, since embryos and ovulation sites can be quantified in the same litters. This is likely, actually, and would increase the paper's impact if true. Can the authors elaborate on the effectiveness of the spindle checkpoint in general, and whether it comes into play at all in these mutants. Additionally, the *Chk2*^{-/-} rescue results are only superficially interpreted. The authors state, "The lack of a complete rescue with the loss of *CHK2* is likely due to the persistence of synapsis defects (open circles; Fig. 4e) and/or reduced *MLH1* focus numbers (Fig. 4e and Supplementary Fig.

5b)." The implications of this statement will not be obvious to everyone. Indeed the latter is probably incorrect, because MLH1 deficiency doesn't lead to elimination of oocytes prior to ovulation.

- The authors use MEIOB foci and H2AX foci to conclude that the Shu mutants form normal numbers of DSBs. Probably true, but a Spo11 oligo assay would be a very convincing orthogonal assay.

- Are their two mutants really the same? They are in a complex together, but they are used interchangeably here, and in certain figures there are differences between the two. What does this imply? If they are not the same, why weren't double mutant experiments performed for the Sws genes as well (combined with Chk2 and Brca2 C term).

Minor issues:

- The abstract fails to reflect the significance and context of this paper. To anyone but an aficionado, the Shu complex and BRCA2 C terminus need an explanation. I realize that there may be a word limit in the abstract, but it is important to craft it so diverse scientists can understand the implications of the work.

- Final sentence in intro, last phrase, is uninformative.

- The use of MLH1 as equivalent for crossovers is a bit misleading; yes; it's believed that MLH1 marks crossovers but to say it is a crossover isn't entirely appropriate - metaphase spreads to detect crossovers would be needed to state this definitely.

- p. 5, the statement that "Unlike later stages, early meiotic prophase cells at leptonea and zygonema are increased in Shu single- and double- mutant mice. " is misleading. I think the authors may mean that the fraction of early meiotic cells, as a proportion of all cells in mutant tubules (which lack postmeiotic stages), is higher than wild type. As stated, it sounds like they have more overall numbers of those cells.

- Abnormal misspelled in Fig 2b, bottom left.

- This statement on page 6 is confusing: "Consistent with defects in DSB repair, mutant spermatocytes at early pachynema display γ H2AX on autosomes, which is even more evident in early

pachytene-like cells with synapsis defects (Supplementary Fig. 4d).” I’m not sure how they distinguish between H2ax staining of DSBs vs synapsis defects.

-Supplemental Figure 3E, should an analysis of chromosomal end-to-end fusions in the mutants be conducted?

-Supplemental Figure 4C: no abnormal cells are quantified in this figure as stated in the legend; only part of the symbol key is present in the figure

-typo page 3, last paragraph “Suprisingly”

Reviewer #2 (Remarks to the Author):

Review on Abreu et al.

NCOMMS-18-04038

The paper by Abreu et al. describes the characterization of knockout (KO) mice for two genes, SWS1 and SWSAP1, which plays a critical role in homologous recombination in somatic cells, in meiotic recombination. SWS1, which forms a complex with SWSAP1, is an orthologue of Shu2/Sws1 proteins in yeasts, which plays a role in homologous recombination and DNA repair, particularly by promoting the assembly of Rad51 protein on single-stranded (ss) DNAs. SWSAP1-SWS1 belongs to a new class of a RAD51 mediator, which may be different from RAD51 paralogs; e.g. RAD51B, -C, -D, XRCC2 and -3. The authors created two independent KO mice for both of the genes by using TALEN-mediated gene editing technology. The mutant mice are not lethal which is clearly different from phenotypes of RAD51 paralogs KO mice with embryonic lethality. SWS1 and SWSAP1 KO mice are both sterile in male and female. Cytological analysis of chromosome spreads from testis and ovary shows big reduction of number of RAD51 and DMC1 foci without affecting the number of MOEB1 foci, which marks early recombination intermediates than RAD51/DMC1. Rescued of the infertility by CHK2 deletion and crossover homeostasis is also analyzed in the SWS1 and SWSAP1 KO mice. Moreover, the authors showed that Brca2 C-terminal regions encoded by exon 27 is important for RAD51/DMC1 assembly in the absence of SWS1/SWSAP1. All of the experiments in the paper were conducted well, and the data are in high quality, very much convincing and presented in a good shape. The results in the paper are of great interest to not only researchers in meiosis and

recombination fields but also to general readers. However, prior to publication, some issues described below should be addressed.

Major points:

1. In title, the authors called SWSAPI-SWS1 complex as Shu complex. Given SWS1 in human and mouse was first discovered as a homologue of fission yeast SWS1 and named by Russel's group (Martin, 2006), it is fair to refer to as SWSAPI-SWS1 complex rather than Shu complex.
2. Although the authors tend to claim that SWSAP1-SWS1 is a mediator for both RAD51 and DMC1, it is known that DMC1-assembly is dependent of RAD51 in budding yeast and Arabidopsis. Thus, DMC1 assembly defect in SWSAP1 and SWS1 KO mice might be due to defective RAD51 assembly. The effect on DMC1 assembly is not direct. This possibility should be pointed out and discussed in some extents. Alternatively, the authors should show an additional evidence that SWSAP1-SWS1 is a mediator for DMC1.
3. Since this is the first time to describe meiotic defects for *Swsap1 Brca2d27/d27* mutant mice, it might be better to show MLH1 focus counts in the mutant (to address whether crossover homeostasis still is functional in this double mutant mice or not as in *Swsap1* or *Sws1* KO mice).
4. Statistics: There are some many multiple comparisons in graphs. The correction should be done with Bonferroni's correction. As a result, some P-values may be affected.

Minor points:

1. One possibility to explain defective RAD51/DMC1 assembly might come from reduced protein expression due to loss of the interacting protein. It would be great if the authors show normal RAD51 and DMC1 proteins levels in the mutant mice by western.
2. Given the reduced focus counts for RAD51/DMC1 (and increased number of MEIOB foci) in the mutant mice, it is also important to check whether the signal intensity of the focus is different in wild-type and the mutant mice such including *Swsap1 Brca2* double.
3. Page 6, second paragraph, line 3: Please add "signal intensity of" between "quantified chromatin-bound".
4. Page 6, second paragraph, line 8-9: the increased number of MEIOB foci does not necessarily come from unstable RAD51/DMC1 binding, rather longer life-span of MEIOB-containing foci possibly due to impairment of downstream events. It is better to rephrase the sentence here.
5. Figure 2b, % of stages: The total % in the mutant mice does not reach 100%, while that in wild type is ~100%. Additional information should be presented in the legend
6. Figure 2c and Supplementary Figure 3d: In mutant mice, there Y-shaped synaptic chromosomes, in which all three parts were stained with both SYCP1 and -3 (triradial synapsis?). Is this typical in the mutant mice? How the authors explain these abnormal pairing in three ways? For

me, it is hard to explain this abnormal synapsis since this structure does not have any unsynaptic chromosomes stained only for SYCP3.

7. Figure 2f and 2g (and 5e and 5g): What is “abnormal cells” in graph? If this comes from Figure 2b (5b), there are no data for focus counts in abnormal class in “early zygonema” (Figure 2b shows that 1/3-1/2 should be abnormal class). Need some explanation.

8. Supplementary Figure 3d, bottom left image: This should contain a blue color for DAPI. It would be great if the authors add the blue or enhance it more (and delete white lines in sws1 swsap1 mouse or show what the line means).

We appreciate the reviewer's comments and suggestions, which have strengthened our paper. A point-by-point response to the reviewers' comments (AU:) is below in blue, and a brief summary of the new data is as follows:

MEIOB counts: Additional mice

Fig. 3b, 5h and Supp. Fig. 5c (previous Supp. Fig. 4): Added spermatocyte MEIOB counts from three additional *Dmc1*^{-/-} mice (total n=4).

MLH1 counts: Additional mice

Fig. 3f: Added spermatocyte MLH1 counts from the following additional mice: two controls (total n=4); two *Sws1*^{-/-} (total n=3), and one *Swsap1*^{-/-} (total n=3).

Fig. 3g: Added MLH1 foci per bivalent from the new mice in Fig. 3f.

Fig. 3i: Added oocyte MLH1 counts from one additional control (total n=2) and two additional *Swsap1*^{-/-} mice (total n=3).

Supp. Fig. 5h: Added additional analysis on the number of MLH1 foci per bivalent from the new mice in Fig. 3i.

***Sws1*^{-/-} *Chk2*^{-/-} double mutants: new analysis**

Fig. 4a and Supp. Fig. 6a (previous Supp. Fig. 5):

Ovary and body weights: eleven controls (total n=16); two *Chk2*^{-/-} (total n=7); five *Sws1*^{-/-} (total n=5); three *Sws1*^{-/-} *Chk2*^{-/-} (total n=3).

Testis weight and body weights: eight controls (total n=16); four *Chk2*^{-/-} (total n=14); four *Sws1*^{-/-} (total n=4); two *Sws1*^{-/-} *Chk2*^{-/-} (total n=2).

Fig. 4d: Primordial follicle counts: three controls (total n=4); two *Chk2*^{-/-} (total n=5); four *Sws1*^{-/-} (total n=4); three *Sws1*^{-/-} *Chk2*^{-/-} (total n=3).

Supp. Fig. 6b: Ovary sections stained for DDX4 from *Chk2*^{-/-}, *Sws1*^{-/-}, *Sws1*^{-/-} *Chk2*^{-/-}; n=3 mouse per genotype.

Supp. Fig. 6d: Ovulated oocytes, number of mice: *Swsap1*^{+/-} *Chk2*^{-/-}, total n=12; *Swsap1*^{-/-} *Chk2*^{-/-}, total n=8.

Implantation sites (sum of embryos and resorbed), number of mice: *Swsap1*^{+/-} *Chk2*^{-/-}, total n=9; *Swsap1*^{-/-} *Chk2*^{-/-}, total n=7.

Supp. Fig. 6f: Added H&E-stained testis sections from control, *Chk2*^{-/-}, *Sws1*^{-/-}, and *Sws1*^{-/-} *Chk2*^{-/-}, n=1 mouse per genotype.

Supp. Table 3c: Fertility assessment of *Sws1*^{+/-} *Chk2*^{-/-} and *Sws1*^{-/-} *Chk2*^{-/-}, n=1 mouse per genotype.

Chromosomal end-to-end fusions analysis: new analysis

Supp. Fig. 3f: Analysis of chromosomal end-to-end fusions in the Shu mutants. Same mice as in Supp. Fig. 3a-e.

RAD51 and DMC1 protein analysis: new analysis

Supp. Fig. 4c, d: Westerns for RAD51 and DMC1 in testis protein extracts. Mice: For RAD51 control, n=4; *Sws1*^{-/-}, n=2; *Swsap1*^{-/-}, n=4; For DMC1 control, n=2; *Sws1*^{-/-}; *Swsap1*^{-/-}, n=1.

Figure move:

Supp. Fig. 3c,d: Moved to panels a,b in a new Supp. Fig. 4; subsequent Supp. Fig. numbers are increased one unit.

Reviewers' comments:

Reviewer #1 (Remarks to the Author):

The authors explore the roles of 2 Shu complex proteins in mammals by generating mouse mutants. It turns out that the only effects are in meiosis. The interest here is that there is an unique phenotypic consequence of mutating two component genes: impaired (but not abolished) loading of RAD51 and DMC1 recombinases onto resected meiotic DSBs. Although the Shu complex has been investigated in other organisms, the roles in mice have distinctions.

In general, the work is extensive and well presented. The story will be of interest, not only because the role of these proteins is elucidated, but also because the unique phenotypes may be useful for other investigations, for example in checkpoint control (as they did here), and the role of sub-optimal genomic integrity in reproduction and embryonic viability (which they developed to a lesser extent). The overall impact is viewed to be above average, as is the quality of this manuscript.

AU: We are pleased that the reviewer is positive.

Major issues:

- a number data points are derived from only 1 mouse, for example *Sws1*^{-/-} in Fig 3e,f; *Dmc1*^{-/-} in Fig 3a,b; *Sws1*^{-/-} and *Swsap1*^{-/-} oocytes in Fig. 3h,i for Mlh1 focus counts. The latter is bad, given the relatively small difference in average foci, and the fact that the mice were not fully inbred (“backcrossed for 3-6 additional generations”); there are known to be genetic differences in Mlh1 foci in different inbred strains. This shortcoming cannot be overcome by simply scoring more spreads from a single mouse.

AU: We addressed this by adding data from additional animals with one exception (see below). The results are comparable to that previously reported with a smaller number of mice.

***Sws1*^{-/-} spermatocytes in Fig 3e,f for MLH1 focus counts:**

We added data from two *Sws1* mutants ($\Delta 1A$ and $\Delta 1G$) for a total of 3 mutants, along with their littermate controls. The mean number of MLH1 foci with the two additional mutants is comparable to that previously reported with the single $\Delta 1A$ mutant (18.0 foci), as is the control mean (23.5 foci).

***Swsap1*^{-/-} spermatocytes in Fig 3e,f for MLH1 focus counts:**

We added data from one additional *Swsap1* mutant for a total of 3 mutants. The mean number of MLH1 foci is nearly identical adding this new mutant to that previously reported for the two mutants (18.4)

***Dmc1*^{-/-} in Fig 3a,b, 5h and Supp. Fig. 5c:**

We counted MEIOB foci from three additional *Dmc1* mice for a total of 4 mutants. The mean number of MEIOB foci is similar to the number that we previously reported with one mouse.

***Sws1*^{-/-} oocytes in Fig. 3h,i for MLH1 focus counts:**

We harvested embryos from 5 pregnant mice and made chromosome spreads from all 17 female embryos obtained from them while the embryos were being genotyped.

Despite this substantial effort, unfortunately none of the female embryos was mutant, precluding us from adding MLH1 counts on *Sws1* oocytes.

However, we have now provided data on the *Chk2* crosses with the *Sws1* mutant in Fig. 4a, Supp. Fig. 6a,b, and Supp. Table 3c. Notably, *Sws1 Chk2* ovaries are rescued substantially in size relative to the *Sws1* mutant, consistent with a DNA damage response in the single *Sws1* mutant, which leads to a severe depletion in oocytes that can be dampened by the loss of CHK2. Further, *Sws1 Chk2* females can give rise to live births, implying that crossover numbers are not impacted enough to preclude live births. Moreover, MLH1 foci in *Swsap1* males and females are similar (18.4 and 17.3, respectively) which is also very similar for that in *Sws1* males (18.0). Thus, it seems likely that *Sws1* oocytes have similar numbers of MLH1 foci. In support of this is the similarity of the *Sws1* and *Swsap1* mutants in many other respects in males and females.

***Swsap1*^{-/-} oocytes in Fig. 3h,i for MLH1 focus counts:**

We added data from two additional *Swsap1* mutants for a total of 3 mutants, along with their littermate controls. The mean number of MLH1 foci is similar adding the two additional mutants to that previously reported with the single mutant (17.3 foci), as is the number for the controls (24.4 foci).

- the basis for the smaller litter size in *Chk2*^{-/-} *Swsap1*^{-/-} animals should be explored. Mice with smaller oocyte pools may ovulate identical numbers (at least in early life), but may suffer from aneuploidy (associated with reduced crossovers), leading to embryonic lethality. The experiment would be relatively easy, since embryos and ovulation sites can be quantified in the same litters. This is likely, actually, and would increase the paper's impact if true. Can the authors elaborate on the effectiveness of the spindle checkpoint in general, and whether it comes into play at all in these mutants.

AU: To address the basis for the smaller litter size in the *Swsap1/Chk2* double mutants, we sacrificed pregnant females at 12.5 dpc (or in a few cases adult virgin mice) and counted corpora lutea in the ovary as a measure of ovulated oocytes and also the total implantation sites in the uterine horns, including normal embryos and resorbed embryos. Both the number of ovulated oocytes and the number of implantation sites are marginally reduced (~20%) in the *Swsap1/Chk2* double mutants, although neither reaches statistical significance. The number of embryos is reduced (38%) and the number of resorbed embryos is increased about 3-fold, both of which reach statistical significance. These results suggest that the embryo lethality is increased, possibly due to reduced crossovers, but that reduced numbers of ovulated oocytes may also contribute. These data are presented in Supp. Fig. 6d.

Additionally, the *Chk2*^{-/-} rescue results are only superficially interpreted. The authors state, "The lack of a complete rescue with the loss of CHK2 is likely due to the persistence of synapsis defects (open circles; Fig. 4e) and/or reduced MLH1 focus numbers (Fig. 4e and Supplementary Fig. 5b)." The implications of this statement will not be obvious to everyone. Indeed the latter is probably incorrect, because MLH1 deficiency doesn't lead to elimination of oocytes prior to ovulation.

AU: We thank the reviewer for this last point and have removed the quoted statement. Furthermore, we have added sentences to further the interpretation: "Thus, CHK2 is critical for oocyte elimination in the *Shu* mutants. The rescue by CHK2 ablation is likely better in the *Shu* mutants than that observed in *Dmc1* mice because

DSB repair is more proficient, consistent with previous evidence that DSB load is a determinant of oocyte elimination (Rinaldi et al 2017).”

- The authors use MEIOB foci and H2AX foci to conclude that the Shu mutants form normal numbers of DSBs. Probably true, but a Spo11 oligo assay would be a very convincing orthogonal assay.

AU: While the SPO11-oligo assay would further support the conclusions that DSB levels are unaffected, we feel that the gH2AX data is sufficient to make this point.

- Are their two mutants really the same? They are in a complex together, but they are used interchangeably here, and in certain figures there are differences between the two. What does this imply? If they are not the same, why weren't double mutant experiments performed for the Sws genes as well (combined with Chk2 and Brca2 C term).

AU: Every indication is that the *Sws1* and *Swsap1* (and double) mutants have similar or identical phenotypes: from gross phenotypes like viability, infertility in both sexes, similar reduction in testis and ovary weights, and arrest stages, to more subtle phenotypes like RAD51, DMC1, MEIOB, MSH4, MLH1 focus reductions. We focused on SWSAP1 initially because it is considered to be a RAD51 paralog and, moreover, double mutant analysis is costly both in effort and funds, especially when one of the mutants is infertile. We certainly appreciate the point that double mutant analysis could uncover unexpected phenotypes that distinguish the two mutants, and thus attempted to generate *Sws1/Chk2* and *Sws1/Brca2 C term* double mutants.

Sws1/Chk2:

We were successful in generating *Sws1/Chk2* double mutant mice and now provide a detailed analysis of them, essentially recapitulating the data we obtained with *Swsap1/Chk2* double mutants. Thus, we observed that there was a substantial rescue of ovary weight in the *Sws1/Chk2* double mutants (Fig. 4a, Supp. Fig. 6a) accompanied by the presence of follicles at different stages of development (Fig. 4d, Supp. Fig. 6b), as well as a rescue of fertility (Supp. Table 3c). In males, testis weights were not rescued (Fig. 4b, Supp. Fig. 6e), but post-meiotic cells were observed in some tubules (elongating spermatids, Supp. Fig. 6f). Thus, *Sws1/Chk2* analysis recapitulates the *Swsap1/Chk2* partial rescue, demonstrating that CHK2 plays a critical role in oocyte death and a minimal, but detectable, role in halting spermatocyte progression.

Sws1/Brca2 C term:

While we were successful with *Sws1/Chk2*, and obtained similar results as with *Swsap1/Chk2*, we have been unsuccessful as yet in obtaining *Sws1/Brca2 C term* double mutants. Given that the additional analysis we have performed in response to the reviewer's comments demonstrates similar phenotypes for both the *Sws1* and *Swsap1* mutants, we expect that the *Sws1/Brca2 C term* double mutants will behave similarly to the *Swsap1/Brca2 C term* double presented in Fig. 5. Still, there is the possibility that it will behave differently, but even if that would turn out to be the case, we believe it would not substantially change any of the central conclusions of the paper. We have modified the text to be careful to note that the results apply specifically to the *Swsap1/Brca2 C term* double mutant, and we have also softened any more general conclusions by referring to an “intact” Shu complex or by using the words “suggest” or “suggesting”.

Minor issues:

- The abstract fails to reflect the significance and context of this paper. To anyone but an aficionado, the Shu complex and BRCA2 C terminus need an explanation. I realize that there may be a word limit in the abstract, but it is important to craft it so diverse scientists can understand the implications of the work.

AU: As suggested, we have modified the abstract to attempt to make it more accessible to diverse scientists.

- Final sentence in intro, last phrase, is uninformative.

AU: As suggested, we have modified this sentence.

- The use of MLH1 as equivalent for crossovers is a bit misleading; yes, it's believed that MLH1 marks crossovers but to say it is a crossover isn't entirely appropriate - metaphase spreads to detect crossovers would be needed to state this definitely.

AU: As suggested, we have specified that MLH1 foci mark most, as opposed to all, crossovers.

p. 5, the statement that "Unlike later stages, early meiotic prophase cells at leptotema and zygotema are increased in Shu single- and double- mutant mice." is misleading. I think the authors may mean that the fraction of early meiotic cells, as a proportion of all cells in mutant tubules (which lack postmeiotic stages), is higher than wild type. As stated, it sounds like they have more overall numbers of those cells.

AU: As suggested, we have clarified this point.

-Abnormal misspelled in Fig 2b, bottom left. AU: This was corrected.

-This statement on page 6 is confusing: "Consistent with defects in DSB repair, mutant spermatocytes at early pachynema display γ H2AX on autosomes, which is even more evident in early pachytene-like cells with synapsis defects (Supplementary Fig. 4d)." I'm not sure how they distinguish between H2ax staining of DSBs vs synapsis defects.

AU: The supplementary figure (now Supp. Fig. 5d) shows that early pachytene cells with full synapsis show γ H2AX on autosomes. We also observe residual γ H2AX on the subset of synapsed chromosomes in the pachytene-like cells, although the γ H2AX on unsynapsed regions is more prominent (as expected from MSUC). As the reviewer notes, for unsynapsed regions we cannot distinguish whether γ H2AX is due to DSB repair or synaptic defects or both, but the signal on synapsed segments is strong evidence of a DSB repair defect. We modified the text to clarify.

-Supplemental Figure 3E, should an analysis of chromosomal end-to-end fusions in the mutants be conducted?

AU: As suggested, we have now included this analysis in Supp. Fig. 3f and referred to this analysis in the last sentence in the text on page 5 and added to the figure legend.

-Supplemental Figure 4C: no abnormal cells are quantified in this figure as stated in the legend; only part of the symbol key is present in the figure

AU: This was corrected. We thank the reviewer for finding this oversight.

-typo page 3, last paragraph "Suprisingly"

AU: This was corrected.

Reviewer #2 (Remarks to the Author):

The paper by Abreu et al. describes the characterization of knockout (KO) mice for two genes, SWS1 and SWSAP1, which plays a critical role in homologous recombination in somatic cells, in meiotic recombination. SWS1, which forms a complex with SWSAP1, is an orthologue of Shu2/Sws1 proteins in yeasts, which plays a role in homologous recombination and DNA repair, particularly by promoting the assembly of Rad51 protein on single-stranded (ss) DNAs. SWSAP1-SWS1 belongs to a new class of a RAD51 mediator, which may be different from RAD51 paralogs; e.g. RAD51B, -C, -D, XRCC2 and -3. The authors created two independent KO mice for both of the genes by using TALEN-mediated gene editing technology. The mutant mice are not lethal which is clearly different from phenotypes of RAD51 paralogs KO mice with embryonic lethality. SWS1 and SWSAP1 KO mice are both sterile in male and female. Cytological analysis of chromosome spreads from testis and ovary shows big reduction of number of RAD51 and DMC1 foci without affecting the number of MOEB1 foci, which marks early recombination intermediates than RAD51/DMC1. Rescued of the infertility by CHK2 deletion and crossover homeostasis is also analyzed in the SWS1 and SWSAP1 KO mice. Moreover, the authors showed that Brca2 C-terminal regions encoded by exon 27 is important for RAD51/DMC1 assembly in the absence of SWS1/SWSAP1. All of the experiments in the paper were conducted well, and the data are in high quality, very much convincing and presented in a good shape. The results in the paper are of great interest to not only researchers in meiosis and recombination fields but also to general readers.

AU: We appreciate the kind words of the reviewer.

However, prior to publication, some issues described below should be addressed.

Major points:

1. In title, the authors called SWSAPI-SWS1 complex as Shu complex. Given SWS1 in human and mouse was first discovered as a homologue of fission yeast SWS1 and named by Russel's group (Martin, 2006), it is fair to refer to as SWSAPI-SWS1 complex rather than Shu complex.

AU: Thank you for pointing this out. We updated the title to specify that the Shu complex is SWS1-SWSAP1. We have also modified the introduction, introducing the mammalian complex first as SWS1-SWSAP1 and then indicating later in the paragraph that it is called the "Shu" complex for simplicity, given the diversity of the composition of the complexes.

2. Although the authors tend to claim that SWSAP1-SWS1 is a mediator for both RAD51 and DMC1, it is known that DMC1-assembly is dependent of RAD51 in budding yeast and Arabidopsis. Thus, DMC1 assembly defect in SWSAP1 and SWS1 KO mice might be due to defective RAD51 assembly. The effect on DMC1 assembly is not direct. This possibility should be pointed out and discussed in some extents. Alternatively, the authors should show an additional evidence that SWSAP1-SWS1 is a mediator for DMC1.

AU: This is a good point. We have extensively modified our comparison of the mouse SWS1-SWAP1 and budding yeast PCSS mutants to include a discussion of this point. As discussed, we still favor a direct mediator function for the mouse complex (see the first full paragraph on p.11).

3. Since this is the first time to describe meiotic defects for *Swsap1 Brca2^{d27/d27}* mutant mice, it might be better to show MLH1 focus counts in the mutant (to address whether crossover homeostasis still is functional in this double mutant mice or not as in *Swsap1* or *Sws1* KO mice).

AU: *Swsap1 Brca2^{d27/d27}* double mutant spermatocytes do not progress to mid-pachynema, that is, they arrest before it would be possible to observe MLH1 foci.

4. Statistics: There are some many multiple comparisons in graphs. The correction should be done with Bonferroni's correction. As a result, some P-values may be affected.

AU: We respectfully disagree. The Bonferroni correction suggested is not a good choice because it is much too conservative (Rothman, 1990; Benjamini and Hochberg, 1995; Perneger, 1998; Gelman et al., 2012). More importantly, however, it is not appropriate to do even a less conservative and more widely used method such as the Benjamini-Hochberg method for controlling false discovery rate (FDR) (Benjamini and Hochberg, 1995). This is because these corrections can only be applied to independent statistical tests, which is not the case here. First, each statistical test involves comparison of a mutant to the same wild-type dataset, i.e., the tests are not independent. Also, many of the comparisons involve subsets of cells (different meiotic stages) from one genotype compared to another genotype; since each genotype is one set of related measurements (coming from the same animals), the tests are again not independent. Second, the mutants were not selected at random, but instead are either multiple knockout alleles of the same gene or mutations in both members of the same functional complex. This experimental setup again violates the assumption of independence of the multiple testing correction methods. The likelihood is vanishingly small that essentially all of these comparisons would show changes in the same direction and of the same magnitude simply because of sampling error. Correction for multiple testing remains a controversial area in statistics (Rothman, 1990; Benjamini and Hochberg, 1995; Perneger, 1998; Gelman et al., 2012). For our experiments, indiscriminate application of these corrections (particularly the overly conservative Bonferroni method) is inappropriate, and in fact worse than no correction at all.

Benjamini, Y., and Hochberg, Y. (1995). Controlling the false discovery rate: a practical and powerful approach to multiple testing. *J Royal Stat Soc Series B* 57, 289-300.

Gelman, A., Hill, J., and Yajima, M. (2012). Why We (Usually) Don't Have to Worry About Multiple Comparisons. *Journal of Research on Educational Effectiveness* 5, 189-211.

Perneger, T.V. (1998). What's wrong with Bonferroni adjustments. *BMJ* 316, 1236-1238.

Rothman, K.J. (1990). No adjustments are needed for multiple comparisons. *Epidemiology* 1, 43-46.

Minor points:

1. One possibility to explain defective RAD51/DMC1 assembly might come from reduced protein expression due to loss of the interacting protein. It would be great if the authors show normal RAD51 and DMC1 proteins levels in the mutant mice by western.

AU: To address this formal possibility, we performed western blot analysis and observed that RAD51 and DMC1 levels were similar in mutants and controls (added as a new figure panel, Supp. Fig. 4c,d).

2. Given the reduced focus counts for RAD51/DMC1 (and increased number of MEIOB foci) in the mutant mice, it is also important to check whether the signal intensity of the focus is different in wild-type and the mutant mice such including Swsap1 Brca2 double.

AU: Visually, there was no clear difference, so we did not investigate further.

3. Page 6, second paragraph, line 3: Please add “signal intensity of” between “quantified chromatin-bound”.

AU: We corrected this.

4. Page 6, second paragraph, line 8-9: the increased number of MEIOB foci does not necessarily come from unstable RAD51/DMC1 binding, rather longer life-span of MEIOB-containing foci possibly due to impairment of downstream events. It is better to rephrase the sentence here.

AU: We deleted this phrase.

5. Figure 2b, % of stages: The total % in the mutant mice does not reach 100%, while that in wild type is ~100%. Additional information should be presented in the legend

AU: This is due to number rounding; the data file provides the exact numbers.

6. Figure 2c and Supplementary Figure 3d: In mutant mice, there Y-shaped synaptic chromosomes, in which all three parts were stained with both SYCP1 and -3 (triradial synapsis?). Is this typical in the mutant mice? How the authors explain these abnormal pairing in three ways? For me, it is hard to explain this abnormal synapsis since this structure does not have any unsynaptic chromosomes stained only for SYCP3.

AU: Many of the Y-shaped synaptic chromosomes do have asynaptic regions; in the cases in which there appears to be full synapsis, three chromosomes are engaged which involves either complete nonhomologous synapsis or homologous synapsis together with nonhomologous synapsis.

7. Figure 2f and 2g (and 5e and 5g): What is “abnormal cells” in graph? If this comes from Figure 2b (5b), there are no data for focus counts in abnormal class in “early zygonema” (Figure 2b shows that 1/3-1/2 should be abnormal class). Need some explanation.

AU: Staging early-zygonema-like cells is difficult without costaining with SYCP1, “abnormal cells” are not distinguished at this stage. We have added this point to the legend.

8. Supplementary Figure 3d, bottom left image: This should contain a blue color for DAPI. It would be great if the authors add the blue or enhance it more (and delete white lines in sws1 swsap1 mouse or show what the line means).

AU: There is some variability in the H1t signal (blue) as cells enter mid-pachynema and begin to express H1t. This particular panel was acquired at the same settings as the others and the signal is visible when zooming in. The white lines separate other cells on the field. We noted this now in Fig. 3d legend, when they first appear.

REVIEWERS' COMMENTS:

Reviewer #1 (Remarks to the Author):

The authors have responded very well to the critiques, adding substantial new data and revisions to the text. The paper is very solid I have no more substantial concerns.

Reviewer #2 (Remarks to the Author):

Review on Abreu et al.
NCOMMS-18-04038A

The authors addressed most of the comments by reviewers in a proper way. It should be accepted by Nature Communication.

REVIEWERS' COMMENTS:

Reviewer #1 (Remarks to the Author):

The authors have responded very well to the critiques, adding substantial new data and revisions to the text. The paper is very solid I have no more substantial concerns.

-thank you

Reviewer #2 (Remarks to the Author):

Review on Abreu et al.
NCOMMS-18-04038A

The authors addressed most of the comments by reviewers in a proper way. It should be accepted by Nature Communication.

-thank you